

# The contributions of various calcifying plankton to the South Atlantic calcium carbonate stock

Anne L. Kruijt[1], Robin van Dijk[2,6], Olivier Sulpis[3], Luc Beaufort[3], Guillaume Lassus[3], Geert-Jan Brummer[4], A. Daniëlle van der Burg[2,7], Ben A. Cala[1,4], Yasmina Ourradi[4], Katja T.C.A. Peijnenburg[2,5], Matthew P. Humphreys[4], Sonia Chaabane[3], Appy Sluijs[1], Jack J. Middelburg[1]

[1] Department of Earth Sciences, Faculty of Geosciences, Utrecht University, The Netherlands
[2] Marine Evolution and Ecology, Naturalis Biodiversity Center, Leiden, the Netherlands
[3] Aix-Marseille Université, CNRS, IRD, INRAE, CEREGE, Aix-en-Provence, France
[4] Department of Ocean Systems, NIOZ Royal Netherlands Institute for Sea Research, PO Box 59, 1790 AB Den Burg (Texel), the Netherlands
[5] Freshwater and Marine Ecology, Institute for Biodiversity and Ecosystem Dynamics (IBED), University of Amsterdam, Amsterdam, the Netherlands
[6] now at: Institute of Marine and Antarctic Studies, University of Tasmania, Hobart, TAS, Australia
[7] now at: Department of Environmental Biology, Institute of Environmental Sciences, Leiden University, Leiden, the Netherlands

*Correspondence to*: Anne Kruijt, annelaurakruijt@hotmail.com

**Abstract.**

Pelagic calcifying plankton play an important role in the marine carbon cycle. However, field studies quantifying the contributions of multiple calcifying plankton groups to particulate inorganic carbon (PIC) stocks and export into the ocean interior are scarce. Most studies target one specific plankton group and adjust their sampling strategy accordingly, hampering comparisons. Furthermore, the literature is strongly biased towards foraminifera and coccolithophores, so aragonite contributions (e.g., gastropods) remain virtually unconstrained. A holistic view is required for future projections of marine carbon cycle changes. Here, we present the contributions of three main calcifying plankton groups - coccolithophores, foraminifera and planktonic gastropods (comprising heteropods and pteropods) - to PIC stocks and fluxes throughout the water column during a sampling campaign in the South Atlantic Ocean. Coccolithophore calcite dominated the depth-integrated PIC standing stock (~80%), followed by aragonite from planktonic gastropods (~17 %) and calcite from foraminifera (~3 %). The estimated production and export of the calcifying plankton largely depend on assumed turnover times and sinking speeds, which both have large uncertainties. Coccolithophores contributed 92% of the produced PIC and from 52 to 99% of the exported PIC, depending on their mode of sinking. Both the production and export of planktonic gastropods was significantly larger than that of foraminifera. Similarity between our results and those from different ocean basins suggests that these patterns are global in nature, implying that not only coccolithophores but also gastropods may be more important PIC producers than foraminifera, challenging a longstanding paradigm.





## 1 Introduction

Calcifying plankton play a crucial role in the global carbon cycle because the calcium carbonate they produce impacts the
ocean's carbonate chemistry and thus atmospheric carbon dioxide (Archer 1996, Sarmiento 2006). After death of the
plankton, their dense shells serve as ballast and facilitate the flux of particulate organic and inorganic carbon (POC and PIC)
to the ocean interior and sediment (Sundquist and Broecker, 1985; Millero, 2007). PIC can occur in different crystal forms.
In the open ocean, the production of the most stable form, calcite, is likely dominated by coccolithophores (haptophyte
algae) followed by the unicellular, heterotrophic planktonic foraminifera (Neukermans et al. 2023; Ziveri et al., 2023). The
more soluble species aragonite, is produced by gastropods, notably pteropods and heteropods. (Buitenhuis, 2019; Sulpis et
al. 2022, Knecht et al. 2023). To quantify the production and ultimately the export and accumulation in ocean sediments of
both $CaCO_3$ species, it is essential to understand the relative contribution of the different plankton groups (Neukermans et al.,
2023; Ziveri et al., 2023). This information is needed to identify the governing factors and in modelling studies aimed at
reconstructing particle sinking fluxes and projecting changes in the carbonate pump (Planchat et al. 2023).

Because the physiologies, ecologies, functions and sizes differ strongly between calcifying plankton groups, they are
typically studied by separate research communities using different methodologies. This complicates quantitative comparison
between different studies and groups. Recently, several databases have been developed, containing abundance data for
foraminifera (FORCIS, Chaabane et al., 2023), pteropods (MAREDAT, Buitenhuis et al. 2013, Bednaršek et al., 2012;
AtlantECO, Vogt et al. 2020 (pteropods being one of several plankton functional types documented in both these databases))
and coccolithophores (CASCADE, De Vries et al., 2024). Compilers of these datasets made large efforts to unify the abundance
data (De Vries et al., 2024, Chaabane et al., 2023). This includes corrections and adjustments to unify data reported in various
units (POC, PIC, $CaCO_3$ or number of specimens; abundances or fluxes) and samples were obtained through different
techniques (e.g. plankton nets and pumps of different mesh sizes, continuous plankton recorders (CPR), sediment traps and
water sample collection and filtration). Development of these databases is an important step towards an understanding of the
spatial and temporal contribution of these different calcifying groups to the oceanic $CaCO_3$ stock, as well as their global
production, export fluxes and burial. They also assist in assessing the effects of changing ocean chemistry on the distribution
of these organisms (Chaabane et al., 2024). Still, the numerous corrections and assumptions required to quantify the relative
proportions of calcite and aragonite production per group based on these datasets lead to poorly constrained estimates and high
uncertainties.

Most global quantifications of relative contributions of planktonic calcifiers to PIC production in the open ocean are
based on sediment trap and sediment data (e.g. Broecker and Clark, 2009; Baumann et al. 2003; Milliman, 1993). This resulted
in the paradigm that foraminifera and coccolithophores both contribute about 50% to the global pelagic $CaCO_3$ export and
sedimentation (Broecker and Clark, 2009), with a limited or negligible role for gastropods. However, aragonite gastropod





shells often dissolve in the water column before deposition and burial in the sediment, meaning that sediment data cannot be used to quantify gastropod export (Dong et al., 2019; Sulpis et al., 2022). Recently, Ziveri et al. (2023) quantified the relative contributions of these groups to the total particulate inorganic carbon (PIC) pool in North Pacific seawater. They found that coccolithophores dominated the standing stock and production of $CaCO_3$, (~79% standing stock, ~86% of total production) followed by pteropods and heteropods (~14 and ~1% standing stock, ~10 and ~0.3% of total production) and with foraminifera accounting for ~6% of the standing stock and ~2 % of total PIC production, challenging the paradigm based on sediment trap and sediment data. However, although the first of its kind, Ziveri et al. (2023) only provided data along a limited transect in the north Pacific, at one moment in time.

Here, we follow up on the work of Ziveri et al. (2023) and provide measurements of coccolithophore, foraminifera, and planktonic gastropod abundance and related PIC concentrations in a different oceanic setting: a highly oligotrophic location in the South East Atlantic. Note that we only considered pteropods and heteropods, the gastropod species that are planktonic all their life; we will refer to them as 'gastropods' in this paper. We provide counts at a species and life-stage level, measured weights of planktonic gastropods, foraminifera and coccolithophores, and individual inorganic-to-organic carbon ratios for two abundant pteropod species. We use our results to reconstruct the PIC stock and PIC export concentration for each group at our study site in the South Atlantic. With those concentrations, we calculate the contribution of each plankton group to PIC production and export, compare this with the estimates of Ziveri et al. (2023) and the various databases, and assess global applicability.

## 2 Materials and methods

The approach taken to produce our estimates of PIC standing stock, production rates and export fluxes consists of the following steps:

1) Sampling at sea: collecting planktonic gastropods, foraminifera and coccolithophores

2) Sample processing: producing counts and mass estimates (g-$CaCO_3$) per sampled depth interval, for each plankton group.

3) Conversions: calculating the PIC concentration (g-PIC m$^{-3}$) in each depth interval, using the mass estimates and volume of water from which was sampled.

4) Integrating PIC concentration over depth to calculate the stock (g-PIC m$^{-2}$), discriminating between living concentrations and 'exportable' or 'dead' shell concentrations.

5) Calculating PIC production rates and export rates for each plankton group. For this we use literature-based estimates of species turnover time or sinking speeds.

Sections 2.1-2.5 describe these steps and the related methodology for each plankton group. The steps are the same for each plankton group, but the methods differ, notably because of size differences (Figure 1). A distinction can be made between



steps 1-3, which are primarily based on direct measurements, and steps 4 and 5, in which we perform calculations that require assumptions and use of literature estimates. For step 5, we performed Monte Carlo simulations to determine the uncertainty related to the eventual estimates. Besides plankton sampling, we performed water chemistry measurements on samples collected at the same location and time as for the plankton samples. The physical and chemical characteristics of the water column can help explain the plankton abundance and vertical distributions of plankton at our site and enable better comparison with studies in different oceanic settings. These physical and chemical water measurements are described in a separate paragraph at the beginning of the results section.



**Figure 1** Flowchart of the steps taken in this study to obtain raw samples, process them to obtain PIC concentrations and finally use these concentrations to produce estimates of the contribution of each plankton type to the production and export of PIC.

## 2.1 Sampling at sea

### 2.1.1 Study location

Data was collected during an austral summer sampling campaign on the RV *Pelagia* in the South Atlantic Ocean, in February 2023 (Figure 2; Table 1). All data presented in this paper were collected within 48 hours at stations 3, 4, 6, and 9. The stations



were less than 2 km apart and water column characteristics were similar. We therefore treat these four stations as representative

of the same environment. The stations are located ~730 km offshore South Africa, just south of the Walvis Ridge. This area is

relatively understudied in terms of plankton research (see figure 2 in Chaabane et al., 2023), so by sampling here we hope to

contribute to the global-scale coverage of plankton ecological data. The Benguela upwelling system was too remote to

influence our study site. Waters at our study location at the time of sampling were low in nutrient concentrations (see section

3.1) so plankton concentrations were expected to be low. Samples from station 39 (further northwest; Figure 2) and stations 6

and 9 were used to reconstruct the PIC/POC ratio of *Limacina bulimoides* and *Heliconoides inflatus*, two abundant and

cosmopolitan pteropod species.

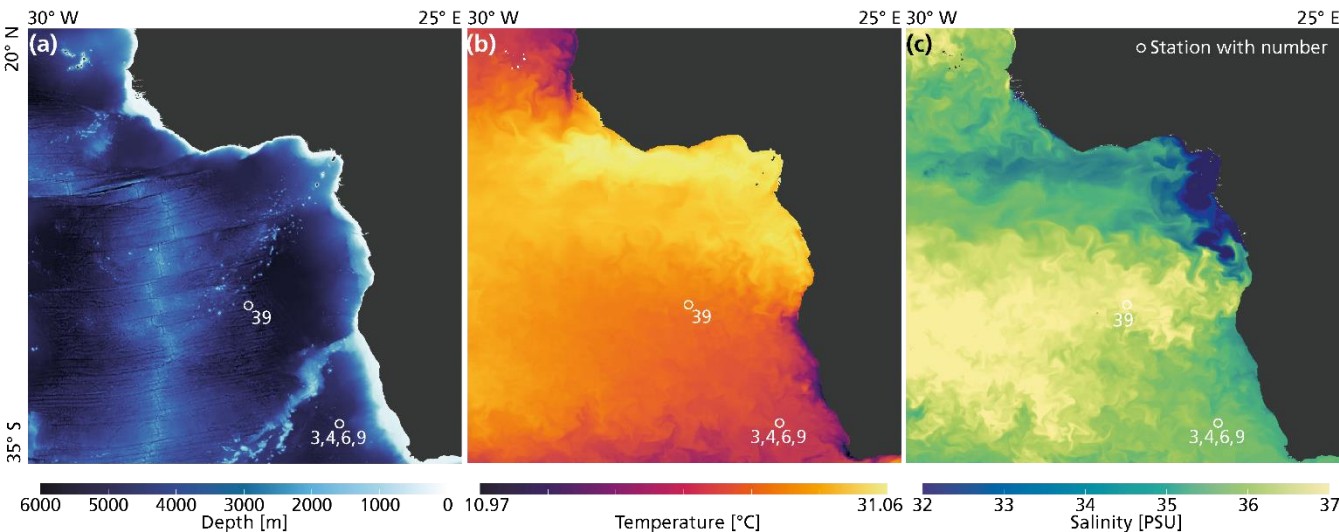

**Figure 2:** The locations of stations 3,4,6, 9 and 39, relative to bathymetry (a), temperature (b) and salinity (c). Surface temperature and

salinity were extracted from the European Union-Copernicus Marine Service (CMEMS) for 18/02/2023 (five days after the day of sampling)

(European Union-Copernicus Marine Service, 2016). Bathymetry data were obtained from the General Bathymetric Chart of the Oceans

(GEBCO Compilation Group, 2022).



| Station | Cast | Date and time start (UTC) | Date and time end (UTC) | Latitude | Longitude | Activity |
|---------|------|---------------------------|-------------------------|----------|-----------|-------------|
| 3 | 1 | 13-2-2023, 11:41 | 13-2-2023, 12:25 | -30.0002 | 9.5005 | CTD rosette |
| 4 | 1 | 13-2-2023, 12:55 | 13-2-2023, 17:27 | -30.0002 | 9.5003 | CTD rosette |
| 6 | 1 | 13-2-2023, 19:25 | 13-2-2023, 21:43 | -30.0005 | 9.5083 | Multinet |
| 9 | 1 | 14-2-2023, 13:58 | 14-2-2023, 16:55 | -30.0152 | 9.5145 | Multinet |
| 39 | 1 | 20-2-2023, 02:41 | 20-2-2023, 05:10 | -15.0305 | -2.0313 | Multinet |

**Table 1:** Location of each station and timing of each sampling activity


### 2.1.2 Data collection

**Planktonic gastropods and foraminifera: plankton tows**

Gastropods and foraminifera were collected with stratified plankton tows (MultiNetR HydroBios "Midi", with an opening of
0.25 m$^2$). This multinet was equipped with five nets made of a 200 µm mesh gauze. Using a stratified net allows for sampling
multiple depth ranges in the water column, the nets each being remotely opened and closed one after the other. Oblique tows
were conducted once at station 6 (after dusk, from 19:25 -21:43 UTC on the 13$^{th}$ of February) and once at station 9 (after noon,
from 13:58 to 16:55 UTC the following day), and the nets dragged at 1-2 knots ship speed. Sampling intervals were
approximately 800-500 m (net 1), 500-300 m (net 2), 300-200 m (net 3), 200-150 m (net 4) and 150 m-surface (net 5), in line
with established methods (Meilland et al. 2021, Peeters and Brummer, 2002). The contents of each net were split using a
Folsom plankton sample splitter. The samples were then rinsed with ethanol, sieved over a 150µm steel sieve, and stored in
96% ethanol at -20°C. Many studies targeted specifically at foraminifera used plankton tows with a mesh size smaller than
200µm (Meilland et al, 2021; Lessa et al. 2020). Using a mesh size larger than the smallest specimens, such as the 200 µm
mesh used in this study, results in biased sampling of foraminifera, underestimating total abundances and skewing species
composition (Chaabane et al., 2024b, Berger 1969; Berger, 1971; Brummer and Kroon, 1988). In fact, Chaabane et al. (2024b)
showed that the 100-200 µm fraction often contains nearly twice as many individuals as the >200 µm fraction. To address this
bias, we used size-normalized catch model equations developed by Chaabane et al. (2024b), to quantify the abundances in the
125-200 µm size fraction (Appendix A). These methods cannot reconstruct the abundances of planktonic foraminifera in the
<125 µm size fraction, which likely contains predominantly juvenile specimens (Schiebel and Hemleben, 2017; Brummer and
Kučera, 2022). Estimating the abundances of very small and rare species remains particularly challenging, and therefore these
data are not interpreted in this study. The reconstructed abundance in the 125-200µm fraction was added to the total count and
the mass of this fraction was estimated using average 125-200 µm shell weights (Appendix A).

**Coccolithophores: water filtrations**

Coccolithophore shells are made up of multiple plate like coccoliths, together creating a spherical cover, termed a coccosphere.
Both intact coccospheres as well as loose coccoliths are too small to sample with a 200 µm multinet. Instead, they were





collected through the filtering of water samples, taken with two rosette casts of Niskin bottles. The casts were given different station names, station 3 and station 4, but are at approximately the same location (Table 1). Samples were taken at 5, 100, 175, 250, 400, 650, (all at station 3), 1000, 2000, 3000, 4000 and 4905 m (all at station 4) (4905 m was at the ocean floor). For each water collection depth, approximately 8 L of sea water were filtered immediately after collection, through a 0.8 µm cellulose
nitrate filter. The filters were then dried overnight at 65°C in the oven and stored at room temperature.

## 2.2 Sample processing: producing counts and mass estimates

### 2.2.1 Planktonic gastropods and foraminifera

**Sorting and counting**

All foraminifera and planktonic gastropods collected with the MultiNets were counted and identified under a microscope (Zeiss SterREO Discovery V.8). Specimens were sorted directly from the multinet samples. Because of storing the samples directly in the freezer after sampling and preserving in ethanol, any body tissue that was present in the shells at the time of sampling was preserved. The presence of body tissue was used to determine the ratio between full and empty shells in each sample. Each sorted multinet sample was checked afterwards by another team member, to minimize counting and identification errors.

Most taxa were identified to species level based on their morphology. Only adult foraminifera were found in the samples, due to the mesh size of the multinet used (200 µm). Planktonic gastropods were classified as juvenile or adult based on morphology and size, using the taxonomy as in the World Register of Marine Species (WoRMS Editorial Board (2025)). For both adult and juvenile gastropods, all but one specimen of the genus *Limacina* were determined on the species level, but several specimens in other genera could only be determined to the genus level. Foraminifera were all determined to species level,

following the taxonomy of Brummer and Kučera (2022). Species smaller than 200 µm were excluded from the dataset since they were most likely caught as bycatch (i.e. entangled in other zooplankton species and therefore not retained by the 200 µm mesh sized net). The sorted specimens were stored in polyethylene, grouped together according to station and net number, species (or sometimes genus) type, organic matter content (full- or empty) and (in case of gastropods) life-stage (juvenile or adult). To determine the PIC and POC content of each net, planktonic gastropod and foraminifera samples were weighed after

sorting, using a high precision microbalance (Sartorius Micro Balance M2P). Most sorted species samples were too small to be weighed individually, so sorted samples were combined into different 'bulk' samples. These bulk samples were grouped by 'net number', 'full specimen' and 'empty specimen', 'adult' and 'juvenile' and 'gastropod' or 'foraminifera'.

**Weighing and ashing**

All bulk samples were dried at 40°C and weighed. Samples were then ashed overnight in a low-temperature asher (LTA) to

remove organic matter and weighed again. The difference between the dried and ashed weight of the samples reflects the weight of the organic matter originally present in the sample (mass organic matter = dried mass – ashed mass). The total number of specimens in the deeper net samples was often so low that the risk of making measurement errors was too large to weigh bulk samples. The PIC content of the unweighed foraminifera samples was reconstructed by multiplying the counts with the average foraminifera PIC weight obtained from the weighed samples. The mass of the unweighed planktonic gastropod





samples was reconstructed using species specific equations for wet and dry weight (Appendix B). The PIC content of those

samples was then calculated using a published PIC/POC ratio for pteropods of 0.27 : 0.73 = 0.34 (Bednaršek et al., 2012). This

ratio has been used in several studies (e.g., Ziveri et al., 2023) to reconstruct pteropod PIC mass.

**Measuring PIC/POC ratio of selected gastropod species**

We strove to use measured rather than calculated PIC mass where possible. The pteropod species *Limacina bulimoides* and

*Heliconoides inflatus*, occurring in high numbers in the surface nets, were not added to the bulk gastropod samples, but weighed

and ashed separately. This was done to reconstruct species specific PIC/POC ratios for these two pteropod species, to be

compared to the ratio provided by Bednaršek et al. (2012) and used in our own study to reconstruct the PIC mass of *L.*

*bulimoides* and *H. inflatus* in the unweighed nets.

**2.2.2 Coccolithophores: filter analysis**

Filters were analyzed through an automated microscope system that can scan filters, recognize the species of each

coccolithophore and estimate its size and thickness. This way, concentrations of each coccolithophore species, as well as the

total concentration of coccoliths and of calcite, can be calculated (see also Appendix C). The original method to recognize and

count the species is described in Beaufort (2004) and the method to estimate the corresponding weights is explained in Beaufort

205    (2021).

**2.3 Conversions: from plankton counts to PIC concentrations**

To obtain PIC concentrations at each sampled depth within the water column, the mass of each plankton type per sampled

depth interval was divided by the volume of water filtered over that interval by the nets (for gastropods and foraminifers) or

by using the filtration system (for coccospheres and coccoliths).

$$(1) \quad PIC\ concentration = \frac{mass\ PIC}{filtered\ volume}$$

A detailed description of all the steps taken to determine the total PIC mass for each plankton type and the relative contribution

of each plankton type to the PIC concentration in the water column can be found in the Supporting Information (Appendix A,

B, C and D).


**2.4 Integrating over depth: from PIC concentrations to standing stock and export concentration**

The productive zone is the depth range where plankton live and calcify. Standing stock is calculated as the integrated

concentration of these living plankton over this productive zone. We assume full shells, e.g. with body tissue inside, within

the productive zone to represent living plankton. Full shells found below the productive zone are assumed to contain dead

plankton. We also calculated the export concentration ($C_{exp}$, mg m$^{-3}$), which refers to the concentration of empty shells or shells

that contain dead specimens. These shells are subject to export, unlike the shells containing living plankton that comprise the

standing stock.



### 2.4.1 Foraminifera

Tell et al. (2022) and Peeters and Brummer (2002) defined the base of the productive zone (BPZ) for foraminifera as the depth
below which shell abundances begin to decline substantially. Most planktonic foraminifer species live in the upper 150 m of
the water column and do not perform diel vertical migration to greater depths (Oberhänsli et al., 1992; Lessa et al., 2020;
Rebotim et al. 2017; Chaabane et al. 2024b). Our shallowest depth interval sampled with the multinet encompassed the entire
upper 150 m of the water column, meaning we do not have information about the variation or trends in shell concentrations
within this range. We did observe a sharp decrease in foraminifera concentrations from the first to the second depth interval
150-200 m) at both stations (see Results section). We therefore consider the peak of production to lie within the upper 150 m
and consider 150 m to be the base of the productive zone. The four multinet samples below the productive zone are considered
to contain only the shells sinking towards the sea floor. Following the Lončarić (2005) and Peeters and Brummer (2002) model,
for a single tow interval the integrated standing stock ($SS_{m2}$, mg m$^{-2}$) can be calculated by

$$(2) \quad SS_{m2} = C_{bpz-0} * (Z_{bpz} - Z_0)$$


where $C_{bpz-0}$ is the measured PIC concentration(mg m$^{-3}$) related to full shells in the tow interval and $Z_{bpz} - Z_0$ is the related
depth range. The export concentration $C_{exp}$ (mg m$^{-3}$) is calculated as:

$$(3) \quad C_{exp} = \frac{MassPIC(empty+full,maxdept\_bpz)}{V_{maxdepth\_bpz}} + \frac{MassPIC(empty,bpz-0)}{V_{bpz-0}}$$


where MassPIC is summed up for all full and empty shells in the depth range below the bpz ($Z_{bpz}$) to the maximum sampling
depth ($Z_{maxdepth}$), $V_{maxdepth-bpz}$ is the total volume of water sampled by all nets below the bpz, and $V_{bpz-0}$ is the filtered volume in
the $Z_{bpz}$-$Z_0$ depth range.

### 2.4.2 Planktonic gastropods

Pteropods and heteropods can actively move through the water column and most species perform significant vertical diel
migration (Lalli and Gilmer, 1989; Wall-Palmer et al. (2018). Commonly reported maximum depths for pteropods are between
200-500 m (Bednaršek et al., 2012), but some studies have found living pteropods as deep as 1000 m (Wormuth, 1981;
Bednaršek et al., 2012). At our study site we found a difference in the depth distribution between station 6 (night) and station
9 (day), with high numbers of full juvenile and adult planktonic gastropod shells in the upper 300 m of the water column at
the daytime station, and planktonic gastropods restricted to the upper 150 m during the nighttime catch (see results section).
This fits with the notion that planktonic gastropods remain closer to the surface at night (Bé and Gilmer, 1977). We therefore
placed the BPZ of planktonic gastropods at 300 m water depth for both stations and calculate $SS_{m2}$ and $C_{exp}$ using Eq. 4 and 5.
$Z_{bpz} - Z_0$ in this case encompasses three tow intervals (nets 5, 4, and 3), so we calculate $C_{bpz-0}$ as total PIC mass in the upper
300 m divided by the total amount of water filtered by the three nets:





(4) $C_{bpz-0} = C_{net3} * V_{net3} + C_{net4} * V_{net4} + C_{net5} * V_{net5}$

Where $C_{net}$ and $V_{net}$ stand for the concentration in a net and the corresponding filtered volume of water.

### 2.4.3 Coccolithophores

For coccolithophores, we assumed the base of the productive layer to be located at the bottom of the deep chlorophyll maximum

(DCM), at 175 m. For coccolithophores we did not make the distinction between full and empty specimens. The integrated standing stock thus comprises all coccosphere mass within the 0-175 m depth range, again calculated using Eq. 4 with $C_{bpz-0}$ being the total coccosphere PIC divided by the total volume of filtered water in the 0-175 m depth interval Export concentration comprises both the sinking coccospheres and coccoliths that sink as part of fecal pellets or marine snow. Filtered water samples from different depths in the water column are unsuited to estimate the sinking flux of coccolithophore-derived calcite. After

filtration, the structure of the larger aggregates, as part of which the coccoliths are sinking, can no longer be observed, as they are fragmented by the filtration. As such, it is difficult to determine the mode of sinking of the coccoliths in the sample. To compare with the planktonic gastropod and foraminifera export concentrations, we calculated the export concentration as the coccolithophore and coccolith mass in the total volume of sampled water in the remainder of the upper 1000 m of the water column. However, which fraction of this coccolithophore-derived calcite was sinking and which fraction was floating without

significant vertical displacement cannot be determined from these samples.

### 2.5 Calculations using literature-based estimates

We used our reconstructed standing stock and export concentrations to provide estimates of the rate with which these calcifying plankton are being produced and the rate at which they are exported to the seafloor after death. For these calculations we used

the average of the standing stock and export concentrations measured at stations 6 and 9. In the absence of directly measured turnover and particle sinking speeds, we had to rely on literature information on the life span of these plankton types and their typical sinking speeds. We performed Monte Carlo simulations using the minimum and maximum estimate for each literature-based parameter value, to assess the uncertainty around the calculated production and export (Supporting Information S2.7). We included a fixed assumed uncertainty of 25% for the measured standing stock and export concentrations, related to potential

errors in the measurements and the assumed integration depth of the standing stock.

### 2.5.1 From standing stock to production

To determine the relative contribution of each of the measured plankton groups to the production of PIC, we needed to make assumptions about the average growth rate of individuals, or the turnover time of the population. For direct comparison of our

results to those of Ziveri et al. (2023), we followed the same approach and calculated the production of PIC as

(5) $PIC\ production = \frac{SS_{m2}}{TT_{pop}}$

where PIC production is in mg C m$^{-2}$ day$^{-1}$, $SS_{m2}$ is the integrated standing stock of the PIC related to the plankton type of interest in mg m$^{-2}$ and $TT_{pop}$ is the average turnover time of the population in days. We calculated the PIC production using



the minimum and maximum turnover times used in the study by Ziveri et al. (2023) as the lower and upper bounds of the
parameter range in the Monte Carlo simulation. The Monte Carlo simulation settings can be found in Table 2.

**2.5.2 From exportable concentration to export flux**

Our plankton net and water filtration samples only provided us with PIC concentrations, not with vertical fluxes. This export
flux however, can be calculated as

(6)  $F_{exp} = C_{exp} * Vs$

with $C_{exp}$ being the measured export concentration of PIC related to a specific plankton species and Vs the sinking speed of
the plankton particle of interest. The minimum and maximum sinking speeds used in the Monte Carlo simulation can be found
in Table 2, together with the reference to the original studies providing the sinking speed estimate. To our notion, sinking
speeds of juvenile planktonic gastropod specimens have never been explicitly determined. Subhas et al. (2023) calculated
sinking velocities of pteropods for a range of shell diameters. We considered the pteropod sinking speeds in the 0.3-0.5 mm
range as determined by Subhas et al. (2023) to be representative of sinking juveniles. We assumed gastropod and foraminifera
shells to sink individually. The export flux was thus calculated by multiplying the concentration of these shells by their
individual sinking rates. This assumption is valid for the relatively large shells of >200 µm that we consider in our study, but
it should be noted that the far smaller juvenile specimens which were not captured by our nets likely sink within marine
aggregates.

The sinking pathway of coccolithophore calcite is complex. Sinking intact coccospheres are relatively uncommon
because the majority of coccospheres are grazed upon by zooplankton and become part of fecal pellets (Ziveri et al. 2023;
Honjo, 1976). Fecal pellets can have high sinking speeds (Table 2, Honjo (1976), Ploug et al. (2008)) and are thought to be
the main pathway through which coccolithophore calcite arrives at the ocean floor. Loose coccoliths have low sinking speeds,
and their export is thus expected to be controlled by the incorporation into sinking aggregates. Loose coccoliths in the photic
zone may dominantly result from shedding by living coccolithophores that are controlling their buoyancy. Loose coccoliths in
the deeper parts of the water column are likely shed from descending fecal pellets (Honjo, 1967). We thus consider three
possible forms in which exportable coccolithophore calcite are present in the water column: as part of a fecal pellet or marine
snow aggregate, as an intact coccosphere or as a loose coccolith. Since our approach does not enable us to determine which
fraction of the sampled coccoliths was part of a fecal pellet, we calculated the export of coccolithophore calcite using three
different modes of sinking: a coccolithophore mode, a loose coccolith mode and a fecal pellet mode (Table 2).



| Plankton group | Planktonic gastropod - adult | Planktonic gastropod - juvenile | Foraminifera > 200 um | Foraminifera 100- 200 um | Coccolith - single | Coccolith - pellet | Coccosphere |
|---|---|---|---|---|---|---|---|
| TT min (days) | 5 | 5 | 14 | 14 | Not relevant | Not relevant | 0.6 |
| TT max (days) | 16 | 16 | 28 | 28 | Not relevant | Not relevant | 10 |
| Vs min (m day$^{-1}$) | 1000 | 450 | 100 | 10 | 0.5 | 50 | 2 |
| Vs max (m day$^{-1}$) | 1900 | 1000 | 500 | 200 | 1.6 | 225 | 6 |
| Reference for Vs | Karakas et al. 2020 | Subhas et al. (2023) | Takahashi and Bé (1984) | Takahashi and Bé (1984) | Honjo (1976) | Honjo (1976), Plough et al. (2008) | Navarro et al. (2018), Bach et al. (2012) |

**Table 2:** Literature based estimates of maximum and minimum turnover times (TT) and sinking speeds (Vs) for each plankton

group, used for the Monte Carlo simulations

### 2.5.3 Reconstructing turnover time

In section 2.5.1 we used our measured standing stock together with literature estimates of turnover times to calculate production rates. The turnover time ($TT_{pop}$) of a plankton population can also be calculated following the approach by Loncaric (2005)

and using measured standing stock and reconstructed export flux:

(7)    $TT_{settl} = \frac{SS_{m2}}{F_{exp}}$

$SS_{m2}$ is the measured integrated standing stock of the adult plankton and the export flux of plankton shells, $F_{exp,}$ calculated using the assumed sinking rate Vs and the measured export concentration $C_{exp}$ of the adult specimen. This approach gives us an estimate of the time needed for the population to completely renew itself, assuming steady state and that all individuals

reach maturity. The method was developed for foraminifera, but we applied it to planktonic gastropods as well. Pteropods and heteropods are still relatively understudied calcifying plankton groups and especially little is known about their life histories and population dynamics (Bednaršek et al. 2016; Manno et al. 2017; Wall-Palmer et al., 2016). We calculated the planktonic gastropods' $TT_{settl}$ separately for juvenile and adult standing stocks and export concentrations, again using Monte Carlo simulations. Export of PIC calculated according to Eq. 8 should at steady state be balanced by *PIC$_{production}$* calculated with Eq.

7. Accordingly, agreement between the export flux $F_{exp}$ and *PIC$_{production}$* would imply that literature community turnover times ($TT_{pop}$) and calculated turnover times with respect to settling ($TT_{settl}$) are internally consistent, while any mismatch would





imply non-steady conditions or bias in either population turnover data or particle settling velocities. Using these two alternative approaches gives us additional insight into the uncertainty around the used estimates.

## 2.6 Water chemistry

### Water sampling and direct measurements

The rosette used to obtain water samples was equipped with conductivity, temperature, depth (CTD) and photosynthetically active radiation (PAR) sensors that directly measured these water column properties during the deployment of the rosette. Water column temperature, chlorophyll-a concentrations and salinity profiles were also obtained using a sensor system mounted on the plankton multinet. Water samples were taken from the Niskin bottles on the rosette, for carbonate system (pH, total alkalinity TA, dissolved inorganic carbon DIC), salinity and nutrient measurements. Carbonate system water samples were collected following the best-practice recommendations (Dickson et al., 2007). If the samples could not be analyzed within 12 hours of collection, they were poisoned with a saturated mercury (II) chloride solution and stored in the dark, for later analysis on land. Samples for macronutrients (ammonia, phosphate, nitrite, nitrate and silicate) were taken using high-density polyethylene syringes (TerumoR) with a three-way valve. The syringe was subsequently used to filter the water through a 0.2 μm AcrodiscR filter and subsamples were transferred into 5 ml polyethylene vials after rinsing each vial three times with the sample before being capped. Macronutrient samples were stored at -20°C, except for those for silicate, which were kept at 4°C, for later analysis on land.

### Lab measurements

Seawater pH was measured on board using the spectrophotometric method of Clayton and Byrne (1993) and Liu et al. (2011). TA and DIC were measured at NIOZ Texel with a VINDTA 3C (Versatile Instrument for the Determination of Total inorganic carbon and titration Alkalinity; no. 14 and 17, Marianda, Germany). The measured samples were calibrated against batch 205 of the certified reference material provided by Andrew G. Dickson (Scripps Institution of Oceanography, USA). Before the TA measurement, DIC was subsampled and subsequently analysed on a QuAAtro Gas Segmented Continuous Flow Analyser (manufactured by SEAL Analytical), following the method described by Stoll et al. (2001). Macronutrient concentrations were also measured with segmented flow spectrophotometric analysis (SEAL QuAAtro instruments) at the laboratory of the NIOZ Texel (Hansen and Koroleff, 1999; Helder and De Vries, 1979; Murphy and Riley, 1962; Strickland and Parsons, 1972). Carbonate ion ($CO_3^{2-}$) and bicarbonate ion ($HCO_3^{-}$) concentrations and aragonite and calcite saturation states were then calculated from TA and pH with PyCO2SYS v1.8.3 (Humphreys et al., 2022).

## 3 Results

We first present water chemistry data to describe the oceanographic setting in which our plankton samples were collected. This is followed by the results of foraminifera and gastropods identification, counting and weighing, including PIC/POC ratios of the abundant planktonic gastropods *H. inflatus* and *L. bulimoides*. We then present the measured coccolithophore and





coccolith abundance. We compare the contribution of the three different calcifying plankton groups to the total PIC stock, and finally we present the living or 'standing' stock (SS), export concentrations, production rates, export fluxes and turnover times related to the different plankton types.

## 3.1 Water column properties

The water column at our study site at the time of sampling represented summer conditions, with stratification into three distinct layers: a well-mixed surface layer from 0-50m, a summer thermocline from approximately 50-100 m and a permanent thermocline stretching from 100 m to a depth of 1000 m. Phosphate and nitrate were depleted in the surface layer and showed subsurface maxima at 1000 m and 1100 m respectively (Figure 3). The deep chlorophyll maximum was located at ~100 m. Carbonate chemistry followed expected patterns (Lauvset, et al., 2024) throughout the water column, co-varying with temperature and salinity and impacted by the biological pump (Middelburg et al., 2020). Alkalinity was highest at the surface and lowest at 650 m depth, in line with the salinity profile. DIC was lowest at the surface and increased with depth, inversely correlated with temperature. As a consequence, the aragonite and calcite saturation horizons were at 900 and 3900 m respectively, indicating that stock assessments were not impacted by dissolution within the productive zone.








**Figure 3 a-o:** Measured physical and chemical water column properties at station 3 and 4.





### 3.2 Planktonic gastropod and foraminifera concentrations

We identified 13 species and 11 genera of planktonic gastropods and 12 species in 6 genera of foraminifera (See Data
availability section). The most abundant gastropods were the pteropod species *H. inflatus* and *L. bulimoides* and the heteropod
genus *Atlanta*, consistent with previous work in the south-east Atlantic Ocean (Burridge et al., 2017). *H. inflatus* and *L.
bulimoides* are found in tropical to subtropical waters around the world (Bé and Gilmer, 1977; Janssen et al., 2019). The most
abundant foraminifera were *Trilobatus sacculifer*, *Globorotalia cultrata* and *Globigerinella siphonifera*. They are species
common to the South Atlantic and reported by previous studies in the same area as our study site (Chaabane et al., 2024b;
Lessa et al. 2020). This gives us confidence that our sampling and counts are representative of plankton community
composition in this area. The total amount of foraminifera and gastropods was higher at station 6 (after dusk) than station 9
(afternoon). The depth distribution of shells also differed between station 6 and station 9, with more full shells deeper in the
water column at station 9 (Figure 4). At both stations, in the upper surface nets (0-150 m, net 5) we found mostly full shells of
adult gastropod and foraminifera. Their concentrations decreased with depth, while the concentration of empty shells increased
slightly (Figure 4). We found not only empty adult shells, but also high concentrations of empty juvenile planktonic gastropods
in our nets, which are part of the export flux. This suggests that many gastropods do not reach maturity.

The measured PIC/POC ratio of adult and juvenile *L. bulimoides* were 0.87 (Standard error = 0.08) and 1.1 (SE = 0.3),
respectively, both much higher than the average 0.37 (0.73 POC : 0.27 PIC, SE = 0.01) presented by Bednaršek et al. (2012).
Juvenile *H. Inflatus* specimens had a PIC/POC ratio of 0.36 (SE = 0.07), closer to the Bednaršek et al. estimate, but the
PIC/POC ratio of the adult specimens was higher (0.50, SE = 0.2) (Figures 5a,b,c,d).





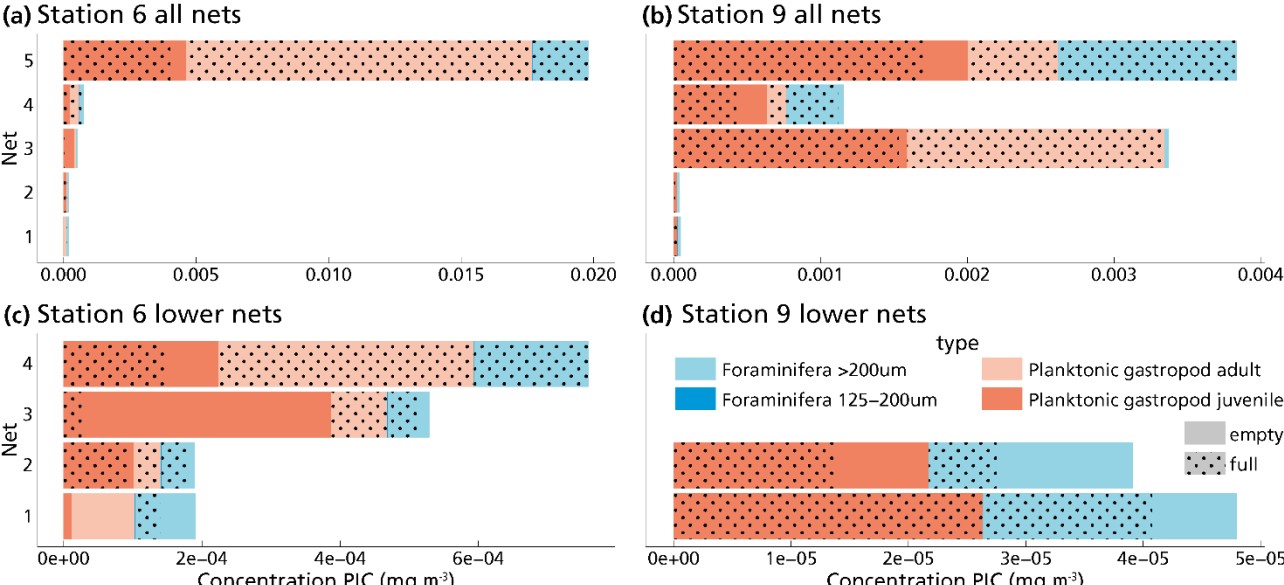

**Figure 4 a,b,c,d:** Measured PIC concentrations of 'small' and 'large' foraminifera and juvenile and adult planktonic gastropods in each net sample at station 6 (Figures a and c ) and 9 (Figures b and d). Figures c and d show the lower nets, with a different scale on the x-axis, to allow for better visualization of the different groups. Note that the concentration of foraminifera in the 125-200 µm size fraction is 2-3 orders of magnitude lower than that of the >200 µm foraminifera, so their contribution to the PIC concentration is hardly visible in these graphs.







**Figure 5 a,b,c,d :** PIC and POC content of the pteropod species *Heliconoides* inflatus and *Limacina* bulimoides; the most abundant pteropods caught with the multinets. Each plot contains three data points, representing samples from three different stations (6, 9 and 39). Each data point is the average PIC/POC ratio of an individual, based on the bulk PIC and POC content of all the H. inflatus and L. bulimoides in the surface nets (net 5) at that station. The regression line, forced through the origin (dashed line), shows the relationship between PIC and POC, for each of the species and life stages. Note that each plot has a different resolution on the x- and y-axis.

### 3.3 Coccolithophore and coccolith concentrations

The highest concentrations of coccospheres were measured at the DCM depth (100 m), below which concentrations dropped to near zero (Figure 6c). The remaining coccospheres found below the DCM peak are interpreted as exported specimens, rather than an *in situ* living community, because of insufficient light levels. A slight increase in coccosphere concentration around 2000m depth, confirmed by visual inspection of the filters containing the sample at this depth, could be a nepheloid layer that contains high concentrations of coccoliths and coccolithophores or fast sinking aggregates trading coccospheres (Beaufort et al. (1999). Different coccolithophore species were identified (see Data availability section), the most abundant being being Emiliania huxleyi, now known as Gephyrocapsa huxleyi. Coccoliths from the most fragile species (e.g. syracosphaera) were found only in the photic zone, and species having a deep photic zone habitat were found at around 175m, but not at greater




depths. This indicates that most of the coccolithophores are found at their living depth. Visual inspection of the samples revealed that deep water samples (>>200m ) contain resistant species with thicker coccoliths (placoliths, helicoliths). The

average measured thickness of the coccolithophores increases with depth ( Appendix Figure G1),  indicating a relative increase in the abundance of thicker species. This could be an indication of more rapid breakup of the thinner species. Our measured coccolith concentrations did not follow the same trend as the coccospheres. The coccolith concentrations in the productive zone (upper 175 m) were about a factor 5-10 larger than coccospheres (Figure 6) and unlike coccospheres, they were present throughout the water column.


**Figure 6 a, b, c:** Concentrations of coccolith (b) and coccosphere (c) PIC plotted next to fluorescence, a measure for relative changes in chlorophyll concentration (a). The fluorescence scale is unitless, since we were not able to calibrate our fluorescence levels with absolute chlorophyll concentrations.  Note the different scales of the coccosphere and coccolith x-axes; coccolith concentrations are one order of

magnitude larger than coccosphere concentrations. The peak of the deep chlorophyll maximum (DCM) is located at 100 m depth, corresponding to the location of the peak in coccosphere concentration, and the bottom of the DCM lies at 175 m.





### 3.4 From counts to standing stock, production and export of calcifying plankton

Coccolithophores calcite dominated the PIC concentration in the top 1000 m of the water column. Coccospheres and loose coccoliths together accounted for 98% of the total PIC concentration measured in the upper 1000 m of the water column. The
PIC derived from gastropods and foraminifera was made up of full and empty shells (Figure 4). PIC concentrations for all species were highest at the DCM and sharply decrease below (Figure 7). The living concentrations of foraminifera and planktonic gastropods were higher than their export concentrations (Table 3), which can be explained by the large sinking speeds of these particles. In contrast, the export concentrations of coccolithophores and coccoliths were a factor 4 higher than the living concentrations of coccolithophores ($C_{living}$ vs $C_{exp}$). This can be explained by the fact that loose coccoliths barely
sink (Honjo, 1976), leading them to accumulate in the water column, until they sink as part of an aggregate.

The integrated coccolithophore standing stock of ~ 7 mg PIC/m$^2$ accounted for 80% of the total standing stock. The average gastropod and foraminifera standing stocks accounted for the remaining 17 and 3%, respectively. In line with this, coccolithophores were by far the largest contributor to the production of PIC (Table 4), accounting for ~92.4% of the total calculated PIC production, followed by 7% by planktonic gastropods and ~0.6% by foraminifera (Table 5, Figure 8). The
relative contributions of the different plankton species to the export flux depends on the assumed sinking mode of the coccolithophore calcite. If we assume that coccoliths and coccospheres sink in isolation, they together contributed approximately 52% of the sinking PIC in the observed water column, followed by 44% from planktonic gastropods and ~4% from foraminifera. If we assume all the coccoliths to be entangled in fecal pellets, meaning they sink faster, they would dominate the export of PIC, contributing 99%.






| Station | Group | Living concentration $C_{living}$ [mg m$^{-3}$] | SSm2 [mg m$^{-2}$] | Export concentration $C_{exp}$ [mg m$^{-3}$] |
|---|---|---|---|---|
| 6 | Planktonic gastropod adult | 0.00574 | 1.72 | 3.92E-05 |
| 6 | Planktonic gastropod juvenile | 0.00178 | 0.533 | 0.000315 |
| 6 | Planktonic gastropod total | 0.00752 | 2.26 | 0.000355 |
| 6 | Foraminifera > 200um | 0.00211 | 0.317 | 6.83E-05 |
| 6 | Foraminifera 125-200 um | 6.32E-07 | 9.48E-05 | 2.05E-08 |
| 6 | Foraminifera total | 0.00211 | 0.317 | 6.84E-05 |
| 9 | Planktonic gastropod adult | 0.000978 | 0.293 | 0 |
| 9 | Planktonic gastropod juvenile | 0.00142 | 0.425 | 0.000108 |
| 9 | Planktonic gastropod total | 0.00240 | 0.718 | 0.000108 |
| 9 | Foraminifera > 200um | 0.00121 | 0.182 | 5.94E-05 |
| 9 | Foraminifera 125-200 um | 3.63E-07 | 5.45E-05 | 1.78E-08 |
| 9 | Foraminifera total | 0.00121 | 0.182 | 5.94E-05 |
| 3,4 | Coccolith | not relevant | not relevant | 0.180 |
| 3,4 | Coccosphere | 0.0395 | 6.91 | 0.00677 |

**Table 3:** Living concentration, integrated standing stock and export concentration of all plankton groups, separated by station, life stage or size (in case of planktonic gastropods and foraminifera) and shape (in case of coccolithophores).






| Plankton group | Planktonic gastropod | Foraminifera | Coccolith - single | Coccolith - pellet | Coccosphere |
|---|---|---|---|---|---|
| Production (mg m$^{-2}$ day$^{-1}$) | 0.157 | 0.0123 | Not relevant | Not relevant | 2.07 |
| stdev | 0.0680 | 0.00400 | Not relevant | Not relevant | 2.05 |
| $F_{exp}$ (mg m$^{-2}$ day$^{-1}$ | 0.182 | 0.0192 | 0.189 | 24.7 | 0.0271 |
| Stdev | 0.0523 | 0.00902 | 0.0755 | 11.2 | 0.0105 |
| production using minimum TT | 0.297 | 0.0178 | | | 11.8 |
| TT calculated (small specimen) | 3.56 | 66.9 | Not relevant | Not relevant | Not relevant |
| Stdev | 2.66 | 89.2 | Not relevant | Not relevant | Not relevant |
| TT calculated (large specimen) | 39.6 | 17.0 | Not relevant | Not relevant | Not relevant |
| Stdev | 32.6 | 17.7 | Not relevant | Not relevant | Not relevant |

**Table 4:** Results of the Monte Carlo simulations for each plankton group. Mean production, export rates and turnover times (for planktonic gastropods and foraminifera only), with their standard deviations (Stdev).





| Plankton group | Planktonic gastropod | Foraminifera | Coccolith (single) + coccosphere (single) | Coccolith (pellet) + coccosphere (single) | Coccosphere |
|---|---|---|---|---|---|
| Production (%) | 7.02 | 0.551 | Not relevant | Not relevant | 92.4 |
| Export scenario 1 (%) | 43.6 | 4.60 | 51.8 | Not relevant | Not relevant |
| Export scenario 2 (%) | 0.73 | 0.077 | Not relevant | 99.2 | Not relevant |

**Table 5:** Relative contribution of each plankton group to the production and export of PIC, based on the mean production and export values calculated using the Monte Carlo simulations (Table 4).





**Figure 7:** The plot shows the measured total PIC concentration derived from coccoliths (b), coccospheres (c), planktonic gastropods (d) and foraminifera (e) next to chlorophyll-a (a) measured at the same location. PIC concentration datapoints for planktonic gastropods and foraminifera only go until 650 m since we sampled only the upper 800 m with the multinet. Coccosphere and coccolith concentrations were obtained all the way to the ocean floor.




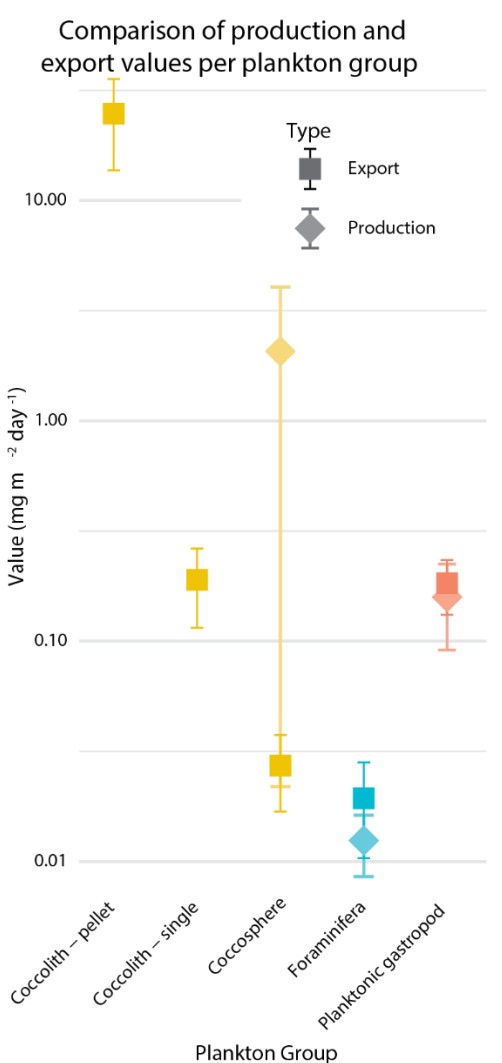

**Figure 8:** Visualisation of the calculated production and export rates listed in Table 4. Error bars show the standard deviations. Values are
plotted on a logarithmic scale, for better comparison between high and low values. The production rate of coccolithophores was calculated
using the coccosphere standing stock, which is why the value is not plotted on the coccolith-pellet and coccolith-single axes; these only
represent sinking material.

## 3.5 Turnover time reconstructed

The average adult planktonic gastropod standing stock and $C_{exp}$ were used in Eq. 9, leading to a calculated $TT_{settl}$ of ~40 days

(Table 4). $TT_{settl}$ based on the juvenile gastropod standing stock and $C_{exp}$ gives us a $TT_{settl}$ of ~3.6 days. These calculations give

us a rough estimate of the turnover time of the gastropod population. The calculated $TT_{settl}$ of the >200µm foraminifer

community is ~ 17 days. The $TT_{settl}$ of the 125-200 µm size range is ~66 days. We refrained from calculating the turnover time

for coccolithophores using the equation by Loncaric and Brummer, because the export flux ($F_{exp}$ in Eq. 9) is too hard to

constrain.






## 4 Discussion

### 4.1 Standing stock and production

Our results show that coccolithophores were the largest contributor to the total PIC concentration and standing stock at the Southern Atlantic Ocean station in February 2023: coccolithophores accounted for ~80% of the PIC standing stock, planktonic gastropods for 17% and foraminifera 3% We realize that one measuring campaign in space and time is not enough to conclude that these results are globally applicable. However, they are in line with the findings of Ziveri et al. (2023), who performed the same type of measurements at five stations along a transect in the North Pacific. Ziveri et al. (2023) found an average contribution of ~79% from coccolithophores, ~15% from gastropods and ~6% from foraminifera across all stations, and ~84, ~12 and ~3% at the two oligotrophic sites in the subtropical gyre. These two subtropical sites are most comparable to our study site in the Southeast Atlantic in terms of ocean chemistry, both located in oligotrophic areas. We did not take direct chlorophyll samples, so our fluorescence measurements (fig, 3, 5, 6) only show the relative changes in chlorophyll concentration through the water column. Satellite data show a value of ~ 0.04 mg m$^{-3}$ at the time of sampling (Appendix Figure E1 and E2), indicating we were sampling in a highly oligotrophic environment. This low value explains our low absolute integrated standing stock values, which for all plankton types are about a factor of 10 lower than standing stocks measured by Ziveri et al. (2023). These low values are however not uncommon for the area; previous research conducted in the proximity of our study site measured integrated foraminifer standing stocks of ~1200 individuals m$^{-2}$ in February 2001 (Table 2.2 in Loncaric et al., 2005) or ~800 individuals m$^{-2}$ (<10 ind m$^{-3}$ in the surface 100 m) in march 2016 (Figure 3, Lessa et al., 2020), which is consistent with the 600 individuals/ m$^{2}$ measured in our study (see Data availability section). Like our measured standing stocks, our calculated relative contributions of plankton groups to the production of PIC are also in line with the previous estimates by Ziveri et al. (2023). They found that coccolithophores contributed 90%, pteropods and heteropods combined 9% and foraminifera 1% to the PIC production at the two most oligotrophic stations along their sampled transect, compared to the ~92.4, ~7% and ~0.6% calculated in our study (Table 5). Both our study and that of Ziveri et al. (2023) agree that coccolithophores are by far the largest contributor to the PIC stock and production in the water column. This strengthens the notion that the dominant role of coccolithophores in PIC stock, followed by planktonic gastropods, is a global phenomenon and is in apparent contrast with the paradigm based on sediments that foraminifera are the second largest contributor to the PIC inventories in the ocean, with a minor role for planktonic gastropods.

Our results appear to deviate from those of Buitenhuis et al. (2019), who compared compilations of biomass observations for coccolithophores, foraminifera and pteropods from the MAREDAT atlas and found that not coccolithophores but pteropods dominate the calcifier biomass in the ocean. However, since their findings are not based on direct measurements of these three calcifiers at the same time and place, their estimated global relative contributions of the different planktonic calcifiers are not necessarily applicable to any real location, e.g. those relative abundances might not be representative of a local ecosystem at any given moment in time. Using database compilations can lead to an unrealistic impression of ocean biology and misinform model parametrization of plankton calcification and should be treated with care. Other issues with the model study by





Buitenhuis et al. (2019) are addressed in more detail in Ziveri et al. (2023). Another database compilation and analysis, by Knecht et al. (2023), used an extended version of the MAREDAT atlas to estimate the global distribution of peteropods and foraminifera biomass. Their results stress the dominance of pteropods over foraminifera in both PIC standing stocks and export, in line with the results presented in our study. We suggest a combination of global scale modelling studies like those of Buitenhuis et al. (2019) and Knecht et al. (2023) and observational work like that of Ziveri et al. (2023) and presented here

will lead to better understanding of plankton abundances on a global scale.

### 4.2 Challenges related to foraminifera and gastropods

The uncertainties in production and export estimates stem from the assumed turnover times and sinking speeds, which vary greatly between species within the plankton groups and in the case of planktonic gastropods are not well established. In our

calculations we tried to make as few further assumptions as possible, by using our measured shell weights to reconstruct the standing stock and related production rates of foraminifera and gastropods. The PIC/POC ratios we measured on *H. inflatus* and *L. bulimoides* were in most cases higher than the estimate from Bednaršek et al. (2012) used by Ziveri et al. (2023) to reconstruct PIC amount. Consequently, aragonite production and export could be underestimated in studies using the Bednaršek estimate, especially when the concentration of planktonic gastropods is high. This uncertainty remains pending

more species- and life-stage-specific PIC/POC ratios for heteropod and pteropod species.

Our calculated production and export rates for both foraminifera and planktonic gastropods roughly balance, with values matching within the uncertainty ranges (Table 4 and Figure 8). To constrain our calculated gastropod and foraminifera export fluxes we used them to reconstruct turnover times ($TT_{settl}$) and compare these to literature-based population turnover times ($TT_{pop}$). The calculated gastropod $TT_{settl}$ range (4-40 days) is wider than the literature-based $TT_{pop}$ estimates (5-16 days)

reported in Ziveri et al. (2023) (Table 2) and the 5-10 days mentioned in e.g. Buitenhuis et al., (2019) and Fabry et al. (1990), but we note that heteropod and pteropod turnover times can vary considerably among species and some studies give far longer estimates (Note, however, that most of this work was done on (sub)polar species, which have longer turnover times, of up to 2 years, e.g: Fabry et al., 1992; Hunt et al. 2008; Wang et al., 2017; Gardner et al. 2023 ). The calculated foraminifera $TT_{settl}$ value for the >200 µm species falls within published estimates of 3-4 weeks (Schiebel & Hemleben, 2017) and the range

reported in Ziveri et al. (2023), but the calculated $TT_{settl}$ for 125-200 µm foraminifera species is slightly longer. Despite the range of $TT_{settl}$ being wider than $TT_{pop}$, all turnover times are of the same order of magnitude, indicating that there is internal consistency between the assumed turnover time ranges used for the production rate calculation and the assumed sinking flux ranges used in the export rate calculation.

### 4.3 Challenges related to coccolithophores

The uncertainty in the sinking mode of coccolithophore calcite complicates comparison among production and export rates of coccolithophore-derived PIC. When assuming all sampled coccoliths were sinking as part of fecal pellets, the calculated export is ~12 times larger than production (Table 4, Figure 8). For production to balance export, this would imply that the export flux

is a pulse-like event, rather than a steady rain (i.e. the steady state assumption that production and export balance does not hold
in the case of a steady rain). Previous research has shown that particle export is indeed highly heterogenous and varies in time
and space (Boyd et al., 2019).

If we assume all coccolithophores and coccoliths to be unattached to fecal pellets and thus sinking very slowly, production
outweighs export by a factor of ~10, indicating that only ~10% of the produced PIC was exported to depth and other processes
were additionally controlling the concentration of coccoliths and coccospheres in the water column. One of these processes
could be the removal of coccolithophore-PIC from the surface ocean through dissolution in the guts of microzooplankton. A
recent study by Dean et al. (2024) showed that 60-80% of the coccolithophore calcite produced in the photic zone dissolves in
the guts of microzooplankton.

An imbalance between our export and production values can also, as for the planktonic gastropods and foraminifera, stem from
the uncertainty related to both sinking speed and turnover time estimates. If we calculate the production of coccolithophore
calcite using the minimum turnover time, production and fecal pellet export rates lie much closer together (Table 4).

We did not have direct means to determine the sinking mode of the coccolithophore-derived PIC and thus can only speculate
about the processes controlling export flux and PIC concentration at our study site. Some insight into the sinking mode can be
obtained through looking at coccolith concentration measurements throughout the entire water column. We measured high
concentrations of coccoliths not only at the surface but all the way to 5000 m depth (fig. 4b). Since single coccoliths have low
sinking rates it is more likely they were exported to these depths as part of aggregates or fecal pellets. We propose that at our
study site, a combination of pulse-like export in the form of aggregates (Turner, 2015), and removal of coccolithophore calcite
by grazing and subsequent dissolution inside microzooplankton guts (Dean et al., 2024) or dissolution due to microbial
respiration induced undersaturation within sinking aggregates, controlled the concentrations of exportable coccolithophore
PIC. This hypothesis fits the observations of Ziveri et al. (2023) who compared measured PIC fluxes with their estimated PIC
production rates and found a ~5 times lower export rate compared to PIC production, which they largely attributed to
dissolution of coccolithophore-derived calcite.


## 4.4 Outlook

Sampling methods such as multinet casts and water filtrations used in this study and sediment traps additionally used in Ziveri
et al. (2023) provide only part of the information needed to understand how plankton sink and in what state they arrive at the
ocean floor. Over recent years, optical particle measurement has emerged as a promising technique to help identify the shape,
size and sinking mode of marine particles (Giering et al., 2020a,b, Trudnowska et al., 2021). Optical devices can be used from
ships, mounted onto a rosette sampler, or installed on autonomous platforms or Argo floats, allowing for large spatial and
temporal coverage. The advantage of these *in situ* imaging techniques is that the particles of interest stay intact and detailed



information can be gained on the shape and size of aggregates carrying, for example, coccoliths towards to ocean interior, and on how these aggregates change with depth. This information cannot be obtained from sediment trap, net or filtered water

samples. However, translating optical signals into flux estimates is challenging, as the density and particle composition cannot be determined from images alone (Giering et al. 2020a). Advances in this field are going fast (Habib et al, 2024; Soviadan et al., 2025) and we suggest that especially combining optical measurements with multinet samplings and sediment traps could provide a holistic picture of the particles being produced and exported from the ocean surface.

Collecting and analyzing particles from the water column remains important to reconstruct particle dissolution.

Dong et al. (2024) used the stable carbon isotopic composition ($\delta^{13}$C) of PIC and POC to identify dissolution and respiration in the water column. Their study however did not provide information on the *in situ* shape and size of the sinking particles. Future studies combining their sampling methods with optical techniques might shed light on both the location of the dissolution in the water column as well as the characteristics of the particles in which this dissolution occurs. Additionally, particle sinking models should be informed with both detailed information on the range of sizes and shapes of marine particles,

as well as the measured $\delta^{13}$C changes and inferred dissolution rates, to further elucidate under which circumstances shallow, respiration-driven dissolution can take place and how this compares to the dissolution within the guts of zooplankton.

Measuring many different parameters at the same time, using a wide range of techniques, is not always feasible of course. In this paper, we articulate that it is essential, however, to quantify the contributions of each of the dominant calcifying plankton groups to PIC production and export separately, instead of just focusing on total PIC, because of their

different fates and preservation potentials. Due to this more comprehensive approach in recent studies, planktonic gastropods are emerging as a previously overlooked but important contributor to PIC production, and the dominant role of coccolithophores in PIC production and export is becoming clearer. More research following the same approach at different locations and moments in time, is required to further constrain the relative contributions of different calcifying plankton groups and understand their patterns and variability through space and time.


## 5 Conclusion

We quantified the relative contribution of three main groups of calcifying plankton to the PIC standing stock at an ocean location in the eastern South Atlantic. Coccolithophores dominated the standing stock of PIC (~80%), with planktonic gastropods accounting for ~17 % and foraminifera contributing only ~3%. These numbers are in line with observations along

a transect in the North Pacific (Ziveri et al., 2023). This consistency suggests that these relative contributions are globally applicable and that the commonly held belief that planktonic gastropods are less important for the PIC stock than foraminifera, should be reconsidered. Production and export rates are hard to estimate based on our multinet and water filter samples alone. Coccospheres and coccoliths clearly dominated the PIC standing stock, but their calculated contribution to the export of PIC towards the ocean interior depended largely on the assumed sinking mode. More integrated research combining imaging

techniques capturing the shape and size of the sinking particles, and sampling techniques enabling chemical analysis, would help better quantify the export of PIC and provide necessary information for models simulating the export of PIC and POC





towards the ocean interior. Finally, we underline the importance of a whole ecosystem approach, rather than focusing on just one of the different calcifying plankton contributing to the PIC stock. This would improve both estimates of current global PIC production and export and predictions of changes in the carbon and carbonate pump.


## Appendix A: Reconstructing foraminifera weights

1) Empty and full shells were picked and counted separately for each net. The full-shell samples from station 6, nets 5,4 were weighed, ashed and weighed again and then divided by the number of shells in the sample at the time of weighing, to obtain average weight of $CaCO_3$ for each shell in those samples. (Table A1).

2) The assumption was made that this average shell weight can be applied to the shells in all net samples. To obtain the total $CaCO_3$ weight of each sample, the original counted number of full and empty shells was multiplied by this average shell $CaCO_3$ weight ($W_{shell}$).

$$(1) \ Total \ CaCO_3 \ mass = count * W_{shell}$$

With $W_{shell} = 0.011016$ mg

3) By using a 200µm mesh size net, a substantial fraction of the foraminifera population was not sampled and the obtained counts are an underestimation of the total foraminifera abundance. The size of this missing fraction was estimated using the method described in Chaabane et al., (2024). This method uses data on the community size structure of foraminifera to obtain multiplication factors by which one size fraction can be normalized to any other size fraction larger than or equal to 125 µm. To scale our measured abundance in the size range 200 µm – infinity ($C^{\{sz\_sup\}}_{\{sz\_inf\}}$) to a theoretical abundance starting at a lower minimum size of 125 µm ($C^{\{\infty\}}_{\{sz\_norm\}}$), we apply equation 3 from Chaabane et al., (2024).

$$(2) \ C^{\infty}_{sz\_norm} = C^{sz\_sup}_{sz\_inf} \frac{f_{max} - f_{sz\_norm}}{f_{sz\_sup} - f_{sz\_inf}}$$

Where sz_norm stands for the normalization size, sz_inf stands for the lower limit of the sampled size fraction and sz_sup stands for the upper limit of the sampled size fraction. Chaabane et al. provided calculated $f_{max}$ values for several sampling depth ranges (Chaabane et al., (2024), table 2). For a sampling depth range of 0-1000 m, an $f_{max}$ of 2.48 can be used. $f_{sz\_norm}$, $f_{sz\_sup}$ and $f_{sz\_inf}$ can be calculated using equation 4 from the same paper:

$$(3) \ f_{sz} = 1 + (f_{max} - 1) * \frac{(Sz - S_{125})}{(Sz - S_{125}) + (S_{half} - S_{125})}$$

taking 125 µm as the normalisation size sz_norm, our used mesh-size of 200 µm as the the lower end of our sampled size range, sz_inf and assuming the upper size limit of our sampled size range, sz_sup, was infinity. The parameter $S_{half}$ was set at 178, again provided by Chaabane et al. (2024) in Table 2 of their paper. This leads to an $f_{sz\_norm}$ of 1, an $f_{sz\_inf}$ of 1.867 and an $f_{sz\_sup}$ of 2.48.

The resulting equation for normalization of our net samples then becomes:



$$(4)\ \left(C^{\{infinity\}}_{\{sz_{norm}\}}\right) = \left(C^{\{sz\_sup\}}_{\{sz\_inf\}}\right) * \frac{(2.48-1)}{(2.48-1.867)}$$

The second term in the equation, the correction factor, is applied to each count result, for every net. For our case the correction factor was 2.4, which means that the measured (counted) abundance largely underrepresents the theoretical abundance. By subtracting the counted abundance from the normalized abundance we then arrive at the theoretical foraminifera abundance in the 125 – 200 µm size fraction, for each net.


4) To obtain the total CaCO₃ weight corresponding to the foraminifera in this missing size fraction, the average weight of foraminifera was estimated by sieving an ashed surface water sample, that was collected at the same place and time using a plankton pump with a 125µm mesh. The sample was sieved over a 200 and a 75 µm mesh to obtain foraminifera in approximately the correct size fraction. 75 foraminifera were then picked and weighed on a high precision microbalance, to obtain the average weight of a small foraminifer.

5) This average weight (step 4) was then multiplied by the calculated number of small specimens (step 3) to obtain the total CaCO₃ weight of the missing fraction.

$$(5)\ CaCO_3 weight\ small\ fraction = (normalized\ abundance - count) * W_{smallshell}$$

With $W_{smallshell} = 2.333\text{E-}06$.

6) PIC concentration was then calculated by summing up all the measured and reconstructed CaCO₃ weights for each net, dividing that CaCO₃ mass by the volume filtered by each net and multiplying that number with 1/8.333; the ratio between the molar mass of carbon and the molar mass of CaCO₃.

| Station | Net | >200 µm Foraminifera PIC concentration: weighed / reconstructed |
|---------|-----|---------------------------------------------------------------|
| 6 | 5 | Weighed |
| 6 | 4 | Weighed |
| 6 | 3 | Reconstructed |
| 6 | 2 | Reconstructed |
| 6 | 1 | Reconstructed |
| 9 | 5 | Reconstructed |
| 9 | 4 | Reconstructed |
| 9 | 3 | Reconstructed |
| 9 | 2 | Reconstructed |
| 9 | 1 | Reconstructed |

**Table A1:** List of samples and the procedure followed to obtain PIC concentrations for the sampled foraminifera.

**Appendix B: Reconstructing planktonic gastropod weights**



1)  Empty and full shells and adult and juvenile shells were picked and counted separately for each net. This way we obtained four planktonic gastropod samples per net: adult-full, adult-empty, juvenile-full and juvenile-empty. The adult full-shell samples from station 6, nets 5,4,3 and station 9, nets 2,3 and the juvenile full shell samples from station 6 net 5,4,2 and station 9, nets 5,4,2,1 were weighed, ashed and weighed again and then divided by the number of shells in the sample at the time of weighing, to obtain average CaCO$_3$ weight for each shell in those nets (see also Table B2).

2)  This average CaCO$_3$ weight per shell was then multiplied by the correct (corrected for splitting the net and corrected for any shells lost after original counting, during the weighing procedure) number of full specimen in the corresponding net.

3)  Two species of pteropods, *L. bulimoides* and *H. inflatus*, were weighed and ashed separately, to reconstruct species- and life-stage specific PIC/POC ratio. For this, the full adult and juvenile specimen from the surface net (net 5) at stations 6, 9 and, additionally, station 39, were used. The shells belonging to one station, one net and one species type and life stage were grouped together and weighed, ashed and weighed again. This number was then divided by the number of weighed shells, to obtain an average organic matter, POC, CaCO$_3$ and PIC weight, to be converted to an average PIC/POC ratio of the individuals at each station (Figure 4 in main text and Table B3).

4)  For the picked samples that where not weighed, the typical PIC weight of each of the planktonic gastropod species present in those net samples (*Atlanta sp., Diacria trispinosa, Creseis sp., Oxygyrus inflatus, Clio pyramidata , Cavolinia sp.*) was calculated using formulas described in Bednaršek et al., (2012) and then multiplied by the count of species of that type present in the sample. This was done for both the full and empty shells present in the net samples. Bednaršek et al., (2012) present three generalized formulas for planktonic gastropod dry weight (DW), each applicable to a typical shell morphology:

For globe shaped specimen :

$$(6) \quad DW = 0.000194 * L^{2.5473} * 0.28$$

For triangular shaped specimen:

$$(7) \quad DW = 0.2152 * L^{2.293} * 0.28$$

For cone shaped specimen:

$$(8) \quad DW = \pi * L^{3*\frac{3}{25}} * 0.28$$

The L in the equation stands for the shell diameter. The factor 0.28 is the conversion from wet weight to dry weight (DW), according to Davis and Wiebe (1985). Dry weight is converted to PIC, POC, mass of CaCO$_3$ and C$_{organics}$ according to the following steps:

$$(9) \quad POC = \frac{DW}{\left(2.5 + 8.333*\left(\frac{0.27}{0.73}\right)\right)}$$

$$(10) \quad PIC = POC * \left(\frac{0.27}{0.73}\right)$$

$$(11) \quad Mass\ CaCO_3 = PIC * 8.333$$





$$(12) \quad Mass \ C_{organics} = POC * 2.5'$$

715 Where 0.27/0.73 is the typical PIC:POC ratio in a pteropod according to Bednaršek et al., (2012), 2.5 is the conversion from POC to $CH_2O$ mass and 8.333 is the conversion from PIC to $CaCO_3$ mass.

Each species was assigned a formula based on its shape and the average DW, POC and PIC of each species was reconstructed (Table B4), using the average shell diameter (L) of the species in question. These shell diameters were measured under a

720 microscope on a few individuals selected manually from the samples.

5) To obtain the total $CaCO_3$ mass in each net sample, the weighed totals (steps 1 and 2), the *H. inflatus* and *L. bulimoides* weights from nets 5 (step 3) and the reconstructed total weight of the unweighed shells (step 4) were summed up . This was done separately for adults and juveniles and full and empty shells, as well as for the bulk total in each net.

725 6) PIC concentration was then calculated by dividing the $CaCO_3$ mass by the volume filtered by each net and multiplying that number by 1/8.333; the ratio between the molar mass of carbon and the molar mass of $CaCO_3$.

| Station | Adult / juvenile sample | Net | Weighed / reconstructed / * |
|---------|------------------------|-----|----------------------------|
| 6 | Adults | 5 | * |
| 6 | Adults | 4 | Weighed |
| 6 | Adults | 3 | Weighed |
| 6 | Adults | 2 | Reconstructed |
| 6 | Adults | 1 | Reconstructed |
| 9 | Adult | 5 | * |
| 9 | Adult | 4 | Weighed |
| 9 | Adult | 3 | Weighed |
| 9 | Adult | 2 | Weighed |
| 9 | Adult | 1 | Weighed |
| 6 | Juvenile | 5 | * |
| 6 | Juvenile | 4 | Weighed |
| 6 | Juvenile | 3 | Reconstructed |
| 6 | Juvenile | 2 | Weighed |
| 6 | Juvenile | 1 | Weighed |
| 9 | Juvenile | 5 | * |
| 9 | Juvenile | 4 | Weighed |



| 9 | Juvenile | 3 | Reconstructed |
| 9 | Juvenile | 2 | Weighed |
| 9 | Juvenile | 1 | Weighed |

**Table B2:** List of samples and the procedure followed to obtain planktonic gastropod PIC concentration. The * indicates that
H. inflatus and L. bulimoides were taken out of the sample before weighing, and weighed separately. Their weights were added
to the total PIC weight afterwards.

| Species | Life stage | PIC [mg ind$^{-1}$] | POC [mg ind$^{-1}$] |
|---|---|---|---|
| *Heliconoides inflatus* | Adult | 0.00974 | 0.0164 |
| *Heliconoides inflatus* | Juvenile | 0.00126 | 0.00315 |
| *Limacina bulimoides* | Adult | 0.0141 | 0.0153 |
| *Limacina bulimoides* | Juvenile | 0.00121 | 0.000738 |

**Table B3:** average measured PIC and POC weights of the species H. inflatus and L. bulimoides.

| Species | Shape | L(average) [mm] | DW [mg] | PIC [mg] | POC [mg] |
|---|---|---|---|---|---|
| *Atlanta sp.* | globe | 0.600 | 1.48 | 9.80 | 2.65 |
| *Diacria trispinosa* | cone | 1.20 | 0.939 | 0.0622 | 0.168 |
| *Creseis sp.* | cone | 1.60 | 1.04 | 0.0690 | 0.187 |
| *Oxygyrus inflatus* | globe | 0.316 | 2.89E-06 | 1.91E-07 | 5.17E-07 |
| *Clio pyramidata* | cone | *No measurements | *0.939 | *0.0622 | *0.168 |
| *Cavolinia sp.* | cone | *No measurements | *0.939 | *0.0622 | *0.168 |

**Table B4:** Planktonic gastropod species that were found in the net samples that were not weighed. The table shows their
assigned shape, average measured diameter and the resulting DW, PIC and POC from equations 6, 8, 9,10,11 and 12.* *Clio
pyramidata* and *Cavolinia sp*. diameters were not measured. Instead, we assume the same average size and shape as for *Diacria
trispinosa*.

**Appendix C: Coccolithophore and coccolith mass calculations**

A 1 cm² section of the nitrocellulose membranes collected during the cruise was mounted between a glass slide and a cover
slip using a UV optical adhesive medium (Norland Optical 74). Each sample was scanned using an automated optical
microscope (Leica DM6000), equipped with a 100× objective lens. Monochromatic blue light ($\lambda = 460 \pm 5$ nm) was used for
illumination. Imaging was carried out with a digital camera (SpotFex, Diagnostic Instruments), capturing 150 contiguous fields
of view (FOVs), each measuring 125 × 125 µm. For each FOV, 14 images were captured at seven different focal planes, with
700 nm steps. Two polarization settings were applied: (1) right circular polarization (RCP) and (2) left circular polarization
(LCP), facilitating the application of the Bidirectional Circular Polarization (BCP) method (Beaufort et al., 2021). The



thickness of the carbonate crystals was determined by combining RCP and LCP images at each focal level using the following equation:


$$(13) \quad d = \frac{\lambda \arctan\left(\left(\frac{LLR}{ILL}\right) - 2\right)}{(\pi\,\Delta n)}$$

where d represents the thickness, $\lambda$ is the wavelength (562 nm), $\Delta n$ is the birefringence of calcite (0.172), and ILR and ILL are the gray values under right and left circular polarizers, respectively. This technique enabled the reconstruction of three-dimensional (3D) images. The seven images from each focal level were stacked using a hyperfocus method to ensure consistent

sharpness across the final 3D image. This configuration achieved a precision of 0.005 µm for thickness and 0.032 pg/µm² for mass (Beaufort et al., 2021). Next, the images were processed using the SYRACO AI software, which integrates morphometry and neural-network-based pattern recognition (e.g. L. Beaufort & Dollfus, 2004). The version used here included a model trained with YOLOv8 on coccosphere images derived from the current FOV collection. SYRACO demonstrated high accuracy in measuring both the mass and length of coccoliths and coccospheres identified in the samples (e.g. L. Beaufort et al., 2022).

Three datasets were obtained: (1) full FOV frames containing images of bright objects, primarily calcite particles, as calcite is one of the few birefringent minerals found in open marine waters, (2) subsets of images that specifically captured the coccospheres present within the FOV and (3) a subsets of images that specifically captured the coccoliths present within the FOV. These images, captured in a way that ensures brightness correlates to thickness, enabled the calculation of the mass of calcite on the membranes, which corresponds to particulate inorganic carbon (PIC), as well as the morphology and mass of the

coccospheres. The calculation of the total $CaCO_3$ mass and the coccosphere (or coccolith) mass (CM) follows these formulas:

$$(14) \quad Total\ CaCO_3\ Mass = Value * MaxThickness * Density * Surface * \frac{FilterArea}{(Liter * Conversion * MaxValue * FieldSurface)} =$$

$$0.0000322 * \frac{Value}{Liter}\ [mg\ L^{-1}]$$

$$(15)\ CM = value * MaxThickness * Density * \frac{Surface}{MaxValue} = value * 3.733\ [pg]$$

Where:

• CM: mass of a coccosphere in picograms (pg)

• Value: sum of all pixel gray levels (representing thickness) within the coccosphere

• Max Thickness: 1.62 µm (at 562 nm)

• Density: calcite density = 2.71 pg µm$^{-3}$

• Surface: area of a pixel = 0.0038 µm²

• Max Value: 256 gray levels (GL)

• Surface Mass in mg cm$^{-2}$

• Conversion = 10: 10 pg µm$^{-2}$ = 1 mg cm$^{-2}$

• FieldSurface = 15950 µm² (126.3 µm x 126.3 µm)

• Filter Area = 1452201204 µm²





•    Liter = number of liters filtered

**Appendix D: Calculating relative contributions of each plankton group to standing stock and PIC concentration**

The relative contribution of each group to the living PIC stock (SS%) is calculated as follows:

-    $SSaverage_g = \frac{\left((SS_{st6,juv}+SS_{st6,adult})+(SS_{st9,juv}+SS_{st9,adult})\right)}{2}$

-    $SSaverage_f = \frac{(SS_{st6,125-200\mu m}+SS_{st6,>200\,\mu m})+(SS_{st9,125-200\,\mu m}+SS_{st9,>200\,\mu m})}{2}$

-    $SSaverage_c = SS_{st3,4,csphere}$

-    $SS\%_{group1} = \frac{SSaverage_{group1}}{(SSaverage_{group1}+SSaverage_{group2}+SSaverage_{group3})} * 100\%$

Where $g$ stands for gastropods, $f$ stands for foraminifera and $c$ stands for coccolithophores.

The average 'total PIC' concentration (so full and empty shells) in the upper 1000 m of the water column is calculated as:

For group = foraminifera or gastropods

-    $PIC_{group,,station,1000m} = \sum_{i=net1}^{net5}\left(\left(PIC_{small,full} + PIC_{large,full} + PIC_{small,empty} + PIC_{large,empty}\right)_i * V_i\right) /$
$\sum_{i=net1}^{net5} V_i$

-    $PIC_{group,average,1000m} = (PIC_{group,station6,1000m} + PIC_{group,station9,1000m})/2$

For group = coccolithophores

-    $PIC_{group,station,1000m} = \sum_{i=0m}^{i=1000\,m}\left(\left(PIC_{coccosphere} + PIC_{coccolith}\right)_i * V_i\right) / \sum_{i=0m}^{i=1000m} V_i$

Where $PIC_i$ stands for PIC concentration at depth i or in net i and $V_i$ stands for the corresponding volume filtered at that depth or with that net. The relative contribution, PIC%, is then calculated as:

-    $PIC\%_{group1} = \frac{PIC_{group1,average,1000m}}{(PIC_{group1,average,100m}+PIC_{group2,average,1000m}+PIC_{group3,average,1000m})} * 100\%$






**Appendix E: Chlorophyll satellite data**

Surface water chlorophyll values for the study area at the time of sampling were downloaded at 4x4km resolution for the 13th of February of 2023 (Global Ocean Colour, CMEMS, 2025).

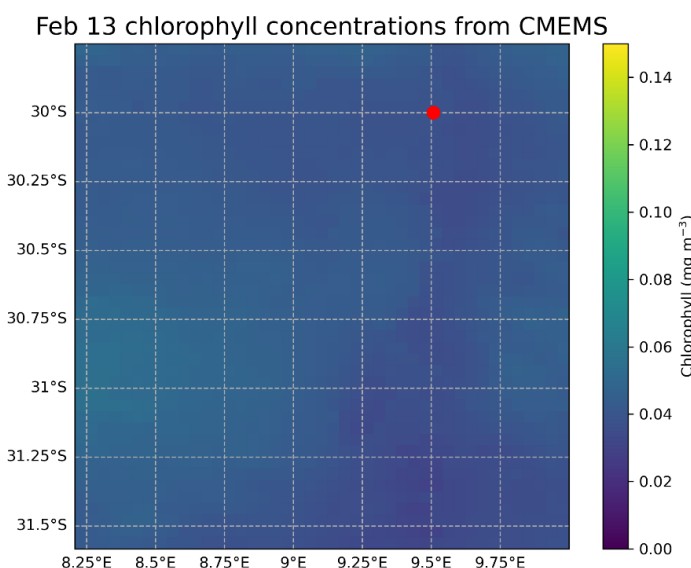

**Figure E1:** Chlorophyll concentrations at the location and time of sampling. The sampling location (stations 3,4,6 and 9) is indicated with a red dot in the map.

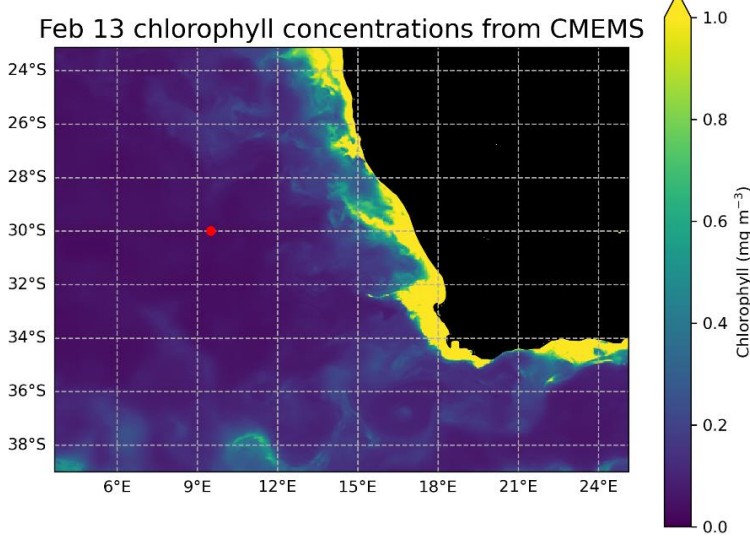

**Figure E2:** Chlorophyll concentrations are plotted for an area stretching all the way to the African coast, to provide some context to the chlorophyll concentrations measured at the sampling location (red dot). The map shows that sampling took place in an oligotrophic area 810 (Chlorophyll concentrations of < 0.1 mg m$^{-3}$.)





## Appendix F: Monte Carlo simulations

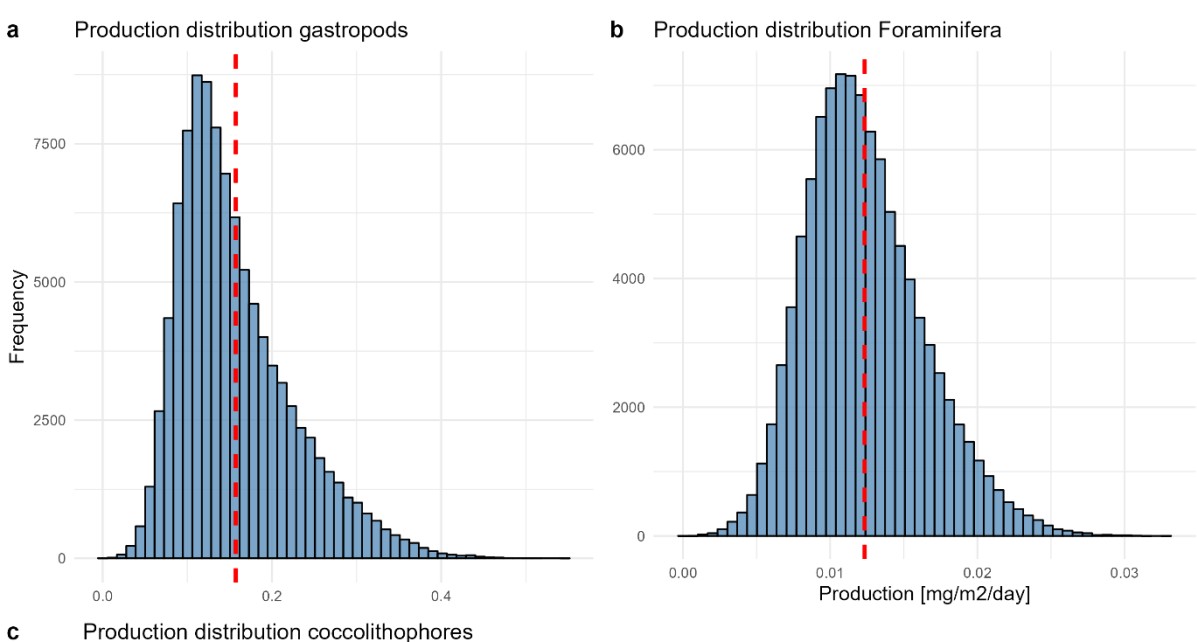

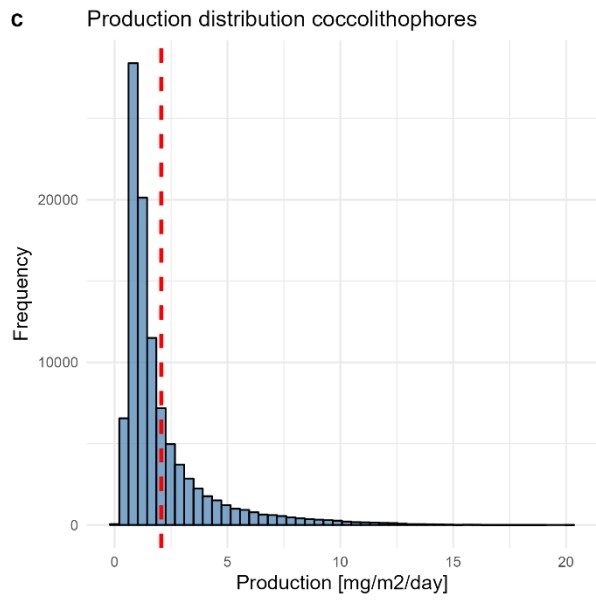

**Figure F1:** Production distribution of the three different plankton types (a, b, c). Red dotted lines show the 95% confidence interval. There are two dotted lines, but they are so close together that they appear as one line on the graph.






**a** F_exp distribution for gastropods

**b** F_exp distribution for foraminifera

**c** F_exp distribution for coccospheres

**d** F_exp distribution for cocco - pellets

**e** F_exp distribution for cocco - single

**Figure F2:** Export flux ($F_{exp}$) distribution of the three different plankton types (a-e). Red dotted lines show the 95% confidence interval. There are two dotted lines, but they are so close together that they appear as one line on the graph.







**Figure F3:** Turnover time distribution of the three different plankton types (a-d). Red dotted lines show the 95% confidence interval. There are two dotted lines, but they are so close together that they appear as one line on the graph. Note the large scale on the x-axis, which is due to the infrequent occurrence of very high maximum calculated turnover times, resulting from the Monte Carlo simulation.



**Appendix G: Coccosphere thickness**

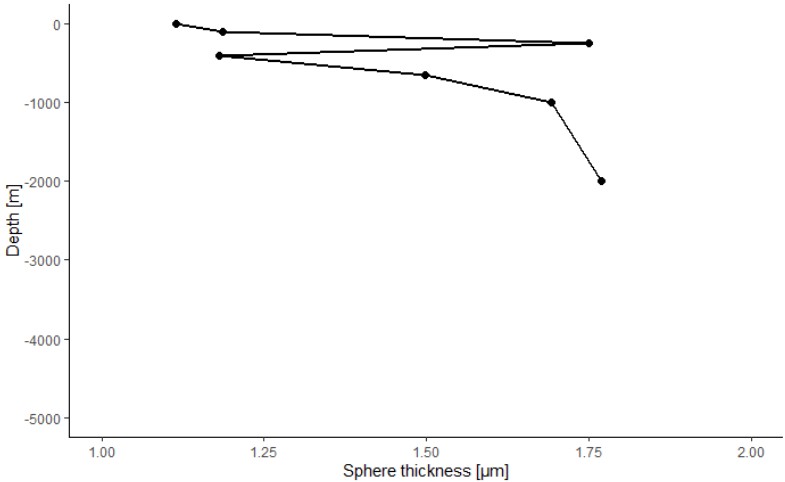

**Figure G1:** Coccosphere thickness with depth in the water column. The plot contains no data points for depths >2000 m, since no coccospheres were found in the filter samples at those depths. Corresponding data can be found in the coccosphere data files made available
on GitHub and Zenodo (Kruijt, 2025).

**Data availability**

All data used in this manuscript has been made available online in our repository on GitHub and Zenodo: https://github.com/AnneKruijt/Calcifying_plankton_paper (Kruijt, 2025). This repository contains: data sets of coccosphere
and coccolith mass at station 3 and 4, coccosphere thickness at station 3 and 4, planktonic gastropod and foraminifera identification and count data from stations 6, 9 and 39, water chemistry data measured at station 3 and 4, excel files with calculations and conversions from raw measurements to PIC concentrations, and the model code (in R) used for analysis (Monte Carlo simulations, standing stock and export concentration calculations and plotting scripts)

Surface water chlorophyll values for the study area at the time of sampling were obtained using E.U. Copernicus Marine
Service Information CMEMS (https://marine.copernicus.eu/) (Figure E1 and E2). The '*Global Ocean Colour (Copernicus-GlobColour)'* dataset was used and chlorophyll values were downloaded at 4x4km resolution for the 13th of February of 2023 (Global Ocean Colour, CMEMS, 2025). Surface temperature and salinity data were extracted from the European Union-Copernicus Marine Service (CMEMS) for 18/02/2023 (five days after the day of sampling) (European Union-Copernicus Marine Service, 2016). Bathymetry data were obtained from the General Bathymetric Chart of the Oceans (GEBCO
Compilation Group, 2022).

**Author contribution**:

Conceptualization of the project was done by AK, OS, AS and JM. Data collection at sea was done by RD, AK, OS, YO, BC and MH. Plankton identification and counting was done by AK, RD, DB, BC, KP and GJ. Coccolithophore data were analyzed



by LB and GL. BC, YO and MH provided the water chemistry measurements. AK and RD were in charge of data curation. AK was in charge of the formal analysis. SC helped applying the size-normalized catch model equations to the dataset. AK wrote the manuscript and all coauthors contributed to the reviewing and editing process. AS and JM were in charge of supervision and funding acquisition.

**Competing interests**: Some authors are members of the editorial board of the journal Biogeosciences.

**Acknowledgements**

We thank the captains and crews of R/V *Pelagia* as well as the NIOZ technicians for their assistance during the BEYΩND cruise. This research was supported by the Netherlands Earth System Science Center funded by the Ministry of Education,
Culture and Science of the Netherlands. AS thanks the European Research Council for Consolidator Grant 771497. We acknowledge the additional support from NIOZ/NWO for funding the BEYΩND cruise.



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
