# Peer review of "The contributions of various calcifying plankton to the South Atlantic calcium carbonate stock"

_EGUsphere, 2025_

## Referee Comment (RC1)

The manuscript "The contributions of various calcifying plankton to the South Atlantic calcium carbonate stock" (<a href="https://doi.org/10.5194/egusphere-2025-4234">https://doi.org/10.5194/egusphere-2025-4234</a>) by Anne Kruijt and colleagues investigates how the three main calcifying plankton groups — coccolithophores, foraminifera, and planktonic gastropods (heteropods and pteropods) — contribute to particulate inorganic carbon (PIC) stocks and export in the South Atlantic Ocean. The authors present data from a sampling campaign, quantifying the depth-integrated PIC standing stock, production, and export for all three groups.

The research is original and addresses a critical question in the field of marine biogeochemistry, providing a more holistic understanding of how the major calcifying plankton groups contribute to carbon cycling and export in the South Atlantic. The study is conceptually strong, clearly written, and presents complex results in a way that is accessible to the reader.

However, certain sections, particularly the *Materials and Methods* section, require clarification and reorganization to improve readability and ensure that methods and assumptions are clearly linked to the presented data. While I understand that the authors use previously published values of turnover time, I see this as highly critical, as it does not take into account the fact that gastropods are, in fact, multicellular organisms that have a more complex life cycle than single-celled foraminifera and coccolithophores (e.g., seasonality, stagnant growth). Therefore, using the same equation for turnover time as for coccolithophores and foraminifera will lead to a drastic overestimation of pteropod contribution, which in fact is seen here (citing from abstract) — "implying that not only coccolithophores but also gastropods may be more important PIC producers than foraminifera." That is the main shortcoming of the paper, in my opinion. I know this method has been used before, but the issue surrounding turnover time needs to be addressed here. Furthermore, the manuscript, while very well written, needs to be checked very carefully for consistency (in used terms, layout, etc.) and formatting of figures needs to improved.

Despite these issues, the manuscript provides valuable insights for the community and is well-positioned to advance our understanding of PIC production and export across multiple plankton groups. I would recommend publication after the authors address the points raised in the review.

I wish the authors success with the revisions and remain available for further feedback.

**Best wishes,**

**Nina Keul**

**Here is a detailed list of comments:**

- **I. 38** "haptophyte": I let the authors be the judge if it is reasonable to assume that the community of BGD all know that term.
  - I. 87: The order in which the three groups are addressed should always be the same; here it is GFC, but in section 2.4 it is F (2.4.1), G (2.4.2), C (2.4.3).

- Fig. 1:
  - "Multinetting" is this indeed a verb?
  - Formatting: in some boxes there is an extra space before the unit (e.g., for PIC), in some not (e.g., counts # for F).
- I. 117: How do you know that the Benguela Current was too remote to influence the study site? Consider rephrasing "remote"
- I. 119: "Samples from station 39 (further northwest; Figure 2) and stations 6 and 9 were used to reconstruct the PIC/POC ratio of Limacina bulimoides and Heliconoides inflatus, two abundant and cosmopolitan pteropod species." This sentence should go to another section.
- **Fig. 2 caption**: The methodology (Copernicus data extraction) should also be described in the text, not just in the figure caption.
- **Fig. 2c**: Station 39 is not readable. What does "Station with number" mean (upper right corner)?
- I. 135: How were station 39 samples collected? Also via oblique multinet tows? Please add.
- **I. 140**: Was this splitting performed on the ship? If yes, how did you ensure a 50:50 split on a moving ship (as the splitter should be level when using)?
- **I. 142**: What about pteropods? How big is the bias here by using a 200 μm net? Do you have estimates? The smallest pteropods we find are usually 80 μm in diameter.
- I. 175: Stored in polyethylene (jars)? Krantz vials? Were they dried before storing? Were they washed (e.g., quick DI bath)?
- **I. 182**: Add a datatable (can be in supplement) to list the n of each category (full, empty, adult, juvenile).
- I. 187: I am a bit puzzled by this comment (in relation to unweighed pteropods). I have weighed individual pteropods before; in the case of *H. inflatus*, the shells of the smaller specimens were still 20+ µg, which could be weighed without problem on an ultramicrobalance.
- **General comment**: Please specify in which institute the analyses were performed (e.g., I. 201 the automated microscope system).
- **I. 222**: "Unlike the shells containing living plankton" rephrase.
- **Equation 4**: You mention that cups were split was that taken into account in the calculations?
- I. 289: The same turnover times as Ziveri et al. are used (5–16 days), which I have a hard time with. While the calculations per se are correct, I feel this is far from reality, where stagnant growth is common, especially in temperate areas as in your study location. Since turnover time is a crucial parameter in flux calculations, using these low turnover times might overestimate pteropod fluxes. Furthermore, in the case of planktonic foraminifera, this is more reasonable; here, values are in agreement with lifespan, as there is no stagnant growth phase for forams.
- Table 2: Add references to all values, not just V.
- I. 330: I value that you try to assess turnover time differently, but again, we know from sediment trap studies that pteropod flux is not the same over the year, so we cannot make these calculations and extrapolate them into a full year, even in a medium-seasonality region such as yours. February, I would imagine, is post-bloom (after summer); I would assume differences in winter. See for instance Oakes et al. 2021, where they only found pteropods in 17/36

sediment trap samples (Oakes RL, Davis CV, and Sessa JA, 2021. Using the Stable Isotopic Composition of Heliconoides inflatus Pteropod Shells to Determine Calcification Depth in the Cariaco Basin. *Front. Mar. Sci.* 7:553104. doi: 10.3389/fmars.2020.55310).

- **Figure 3**: Should be prepared more carefully. For example, the X-axis line is barely visible, and the second Y-axis line is shown incorrectly.
- I. 398 / 299: Does this apply to F and P, or only P? How was full versus empty assessed in foraminifera (staining?)? In Materials and Methods, it gives the impression that only P were assessed in this regard (species or sometimes genus type, organic matter content (full or empty), and in the case of gastropods, life-stage (juvenile or adult)).
- I. 402: Add (SE) after "Standard error = 0.08."
- **Fig. 5**: I am not sure whether this needs to be plotted or can be better represented in a datatable. If the authors decide to keep the figure, its visual appeal needs improvement (remove helper lines, format consistently, include corresponding p-values).
- Section 3.3: Species names in italics; Syracosphaera needs capitalization.
- **Fig. 6**: Remove grid lines; make lines and symbols the same size and thickness.
- **Table 3**: Explain acronyms (SS); check digits and only list significant amounts based on the error associated with the calculations/initial measurements. If in doubt, perform error propagation.
- Tables (general): Check for consistency, e.g., stdev vs Stdev; check the number of digits (see comment for Table 3); apply consistently across manuscript, tables, and figures.
- **Table 5**: Add explanation of export scenario 1 and 2 in the caption.
- **Figure 7**: Same comment as Figure 6. Chlorophyll-a and chlorophyll are used interchangeably; check consistency throughout manuscript.
- Fig. 8: Was, I assume, meant to have a Y-axis.
- **I. 507 / 508**: m² versus m⁻² (consistency); check throughout manuscript, tables, and figures.
- I. 507: Capitalize March.
- **I. 514–516**: This shows that turnover time is vastly overestimated in the case of pteropods.
- I. 667: What is an ashed surface water sample?
- Captions of tables in appendix (B2, B3): Species names need italics.
- **I. 756**: pg μm-2 instead of /μm.

---

## Author Comment (AC1)

Reviewer Nina Keul:

The manuscript "The contributions of various calcifying plankton to the South Atlantic calcium carbonate stock" (https://doi.org/10.5194/egusphere-2025-4234) by Anne Kruijt and colleagues investigates how the three main calcifying plankton groups — coccolithophores, foraminifera, and planktonic gastropods (heteropods and pteropods) — contribute to particulate inorganic carbon (PIC) stocks and export in the South Atlantic Ocean. The authors present data from a sampling campaign, quantifying the depth-integrated PIC standing stock, production, and export for all three groups.

The research is original and addresses a critical question in the field of marine biogeochemistry, providing a more holistic understanding of how the major calcifying plankton groups contribute to carbon cycling and export in the South Atlantic. The study is conceptually strong, clearly written, and presents complex results in a way that is accessible to the reader.

*Reply: We thank Dr. Keul for her compliments and her critical but constructive review.*

However, certain sections, particularly the Materials and Methods section, require clarification and reorganization to improve readability and ensure that methods and assumptions are clearly linked to the presented data. While I understand that the authors use previously published values of turnover time, I see this as highly critical, as it does not take into account the fact that gastropods are, in fact, multicellular organisms that have a more complex life cycle than single-celled foraminifera and coccolithophores (e.g., seasonality, stagnant growth). Therefore, using the same equation for turnover time as for coccolithophores and foraminifera will lead to a drastic overestimation of pteropod contribution, which in fact is seen here (citing from abstract) — "implying that not only coccolithophores but also gastropods may be more important PIC producers than foraminifera." That is the main shortcoming of the paper, in my opinion. I know this method has been used before, but the issue surrounding turnover time needs to be addressed here.

*Reply: We agree that much uncertainty regards the lifestyle of gastropods. We will include text outlining their more complex ecology, including aspects of seasonality and stagnant growth, although constraints on such aspects are insufficient to adapt our general approach. To accommodate this comment by the reviewer, we will adapt our estimate of turnover time, which in our approach is the tuneable parameter. We originally limited our analyses to the approach presented in Ziveri et al. (2023), which enabled us to directly compare our results with theirs. For the revised version we will include calculations using longer turnover time estimates from the literature, resulting in a range of plausible pteropod contributions.*

Furthermore, the manuscript, while very well written, needs to be checked very carefully for consistency (in used terms, layout, etc.) and formatting of figures needs to improved.

*Reply: We thank Dr. Keul for her compliments and critical reading. We will check the revised manuscript very carefully and correct all remaining errors and inconsistencies.*

Despite these issues, the manuscript provides valuable insights for the community

and is well-positioned to advance our understanding of PIC production and export across multiple plankton groups. I would recommend publication after the authors address the points raised in the review.

I wish the authors success with the revisions and remain available for further feedback.

Best wishes,
Nina Keul

Here is a detailed list of comments:

*Reply: A numbered reply is added to each comment.*

l. 38 "haptophyte": I let the authors be the judge if it is reasonable to assume that the community of BGD all know that term.
   1) *We understand the concern. We will add this reference: Jordan, 2009. (https://doi.org/10.1016/B978-012373944-5.00249-2 ), so that readers who do not know the term can find more information.*

• l. 87: The order in which the three groups are addressed should always be the same; here it is GFC, but in section 2.4 it is F (2.4.1), G (2.4.2), C (2.4.3).
   2) *We will change this to the order in section 2.4 (f, g, c).*

• Fig. 1:
o "Multinetting" — is this indeed a verb?
   3) *We will change it to 'MultiNet sampling'.*

o Formatting: in some boxes there is an extra space before the unit (e.g., for PIC), in some not (e.g., counts # for F).
   4) *We will check and correct this.*

• l. 117: How do you know that the Benguela Current was too remote to influence the study site? Consider rephrasing "remote"
   5) *We will rephrase to: 'Waters at our study location at the time of sampling were low in nutrient concentrations (see section 3.1) so plankton concentrations were expected to be low. Our study location is assumed to lie outside the reach of the Benguela upwelling system. The extent of this system is commonly reported to reach only approximately 100 – 200 m offshore (Siddiqui et al., 2023; Hagen et al., 2001; Lutjeharms et al., 1987) although we do note that filaments shedding off the boundary current can reach much further offshore (Rogerson et al., 2025; Lutjeharms et al. 1987).*

• l. 119: "Samples from station 39 (further northwest; Figure 2) and stations 6 and 9 were used to reconstruct the PIC/POC ratio of Limacina bulimoides and Heliconoides inflatus, two abundant and cosmopolitan pteropod species." This sentence should go to another section.
   6) *We agree but also need to mention station 39 here. We shall proceed as follows:*
      • *We will tackle this in the second sentence:*
         o *'All data presented in this paper were collected within 48 hours at stations 3, 4, 6, and 9. The stations were less than 2 km apart and water column characteristics were similar. An exception is station 39, located further north (Figure 2). Data collected at this station are not included in the main analysis of this paper, but will be addressed in section 2.2.1 and Appendix B.'*
      • *In section 2.2.1 we will add a line explaining that samples from station 39, station 6 and station 9 were used to reconstruct the PIC/POC ratio of Limacina bulimoides and Heliconoides inflatus, and explain why we included station 39 (in short: we assumed no latitude related difference in the Limacina bulimoides and Heliconoides inflatus PIC/POC ratios, so in order to obtain a higher number of measured individuals we added the specimen from station 39 to our analysis.)*

• Fig. 2 caption: The methodology (Copernicus data extraction) should also be

described in the text, not just in the figure caption.

*7) We will include the data extraction in the main text.*

• Fig. 2c: Station 39 is not readable. What does "Station with number" mean (upper right corner)?

*8) Agreed, we will change the accompanying text and the colour of the points and numbers in the map.*

• l. 135: How were station 39 samples collected? Also via oblique multinet tows? Please add.

*9) Station 39 samples were collected in the same manner as samples at station 6 and 9. We will clarify this in the text.*

• l. 140: Was this splitting performed on the ship? If yes, how did you ensure a 50:50 split on a moving ship (as the splitter should be level when using)?

*10) Splitting was indeed performed on the ship but weather conditions were good enough to keep the splitter level. This is now included in the manuscript.*

• l. 142: What about pteropods? How big is the bias here by using a 200 μm net? Do you have estimates? The smallest pteropods we find are usually 80 μm in diameter.

*11) We used a 200 μm mesh for several reasons. First, we had to prevent reported clogging and breaking of nets with smaller mesh-sizes and active escape of larger pteropods from fine meshed nets towed at a low speed. This would lead to undersampling of adult pteropods. Moreover, our 100 μm nets broke during a stormy haul and two of the five nets were severely damaged. So ultimately, we use a 200 μm mesh based on the study by Bednaršek et al. (2012), who report that the peak of the biomass lies around 300 μm. However, we realise many studies report that small pteropods are undersampled when using these nets (Bednaršek et al., 2012; Manno et al. 2017; Anglada-Ortiz et al., 2021). We will address this in the revised manuscript as follows:*

- *In the methods section, we will mention that smaller pteropods, especially juveniles, will also be underrepresented by using a 200 μm mesh, and that our pteropod biomass estimates should be interpreted as a conservative estimate.*
- *We will come back to this in the discussion section and point out that the planktonic gastropod biomass is likely higher than our results suggest.*

• l. 175: Stored in polyethylene (jars)? Krantz vials? Were they dried before storing? Were they washed (e.g., quick DI bath)?

*12) These lines refer to the way samples were stored after sorting. We stored them in 1 mL polyethylene vials, in 96% ethanol. We will add this information to the manuscript.*

• l. 182: Add a datatable (can be in supplement) to list the n of each category (full, empty, adult, juvenile).

*13) We made our complete dataset, containing counts for each identified species and category, available on Zenodo as well as a datasheet containing the calculations and conversions from counts and measured weights to PIC concentrations.*

• l. 187: I am a bit puzzled by this comment (in relation to unweighed

pteropods). I have weighed individual pteropods before; in the case of H. inflatus, the shells of the smaller specimens were still 20+ μg, which could be weighed without problem on an ultramicrobalance.

*14) In case of very low yields, we did not want to risk losing (juvenile) individuals while transferring sample from vial to petri-dish to weighing cups. We therefore chose to not attempt at weighing these samples, but instead use the counts, and reconstruct the weights.*

• General comment: Please specify in which institute the analyses were performed (e.g., l. 201 — the automated microscope system).

*15) We will add the institutes for each of the analyses*

• l. 222: "Unlike the shells containing living plankton" — rephrase.

*16) This will be changed to:*

*'We also calculated the export concentration (Cexp, mg m-3), which refers to the concentration of empty shells or shells that contain dead specimens.'*

• Equation 4: You mention that cups were split — was that taken into account in the calculations?

*17) It was. We will add a short line here to clarify this.*

• l. 289: The same turnover times as Ziveri et al. are used (5–16 days), which I have a hard time with. While the calculations per se are correct, I feel this is far from reality, where stagnant growth is common, especially in temperate areas as in your study location. Since turnover time is a crucial parameter in flux calculations, using these low turnover times might overestimate pteropod fluxes. Furthermore, in the case of planktonic foraminifera, this is more reasonable; here, values are in agreement with lifespan, as there is no stagnant growth phase for forams.

*18) We agree that much uncertainty regards the lifestyle of gastropods. We will include text outlining their more complex ecology, including aspects of seasonality and stagnant growth, although constraints on such aspects are insufficient to adapt our general approach. To accommodate this comment by the reviewer, we will adapt our estimate of turnover time, which in our approach is the tuneable parameter. We originally limited our analyses to the approach presented in Ziveri et al. 2023, which enabled us to directly compare our results with theirs. For the revised version we will include calculations using longer turnover times, resulting in a range of plausible pteropod contributions.*

• Table 2: Add references to all values, not just V.

*19) We will add a row to the table containing the references for the turnover time estimates.*

• l. 330: I value that you try to assess turnover time differently, but again, we know from sediment trap studies that pteropod flux is not the same over the year, so we cannot make these calculations and extrapolate them into a full year, even in a medium-seasonality region such as yours. February, I would imagine, is post-bloom (after summer); I would assume differences in winter. See for instance Oakes et al. 2021, where they only found pteropods in 17/36 sediment trap samples (Oakes RL, Davis CV, and Sessa JA, 2021. Using the

Stable Isotopic Composition of Heliconoides inflatus Pteropod Shells to
Determine Calcification Depth in the Cariaco Basin. Front. Mar. Sci. 7:553104.
doi: 10.3389/fmars.2020.55310).

*20) In lines 329-332, we stressed that our calculations are based on the steady state assumption. For any organism, particularly gastropods, this may not be valid. This introduces some uncertainty but in the absence of detailed time series, taking the simplest assumption here is the best solution.*

• Figure 3: Should be prepared more carefully. For example, the X-axis line is
barely visible, and the second Y-axis line is shown incorrectly.

*21) This will be fixed.*

• l. 398 / 299: Does this apply to F and P, or only P? How was full versus empty
assessed in foraminifera (staining?)? In Materials and Methods, it gives the
impression that only P were assessed in this regard (species or sometimes
genus type, organic matter content (full or empty), and in the case of
gastropods, life-stage (juvenile or adult)).

*22) Will clarify the Materials and Methods section 2.2.1. To answer your question, we assed full versus empty for both foraminifera and gastropods.*

- *For gastropods, we could see clearly through the microscope weather a shell contained body tissue or not. If it contained body tissue, it was regarded as 'full', if not, it was regarded as 'empty'*
- *For foraminifera, the same approach was taken. We classified foraminifera tests as 'full' when there was a significant amount of white/green tissue visible within the shell.*

*All foraminifera in our samples, as well as the foraminifera in the reconstructed 125-200 µg size fraction were considered 'adult'. The distinction between adult and juvenile was thus only made for the gastropods.*

• l. 402: Add (SE) after "Standard error = 0.08."

*23) Agreed; we will add this.*

• Fig. 5: I am not sure whether this needs to be plotted or can be better
represented in a datatable. If the authors decide to keep the figure, its visual
appeal needs improvement (remove helper lines, format consistently, include
corresponding p-values).

*24) As suggested, we will change this figure into a table containing, for H. inflatus adult and juvenile and L. bulimoides adult and juvenile:*

- *the size of each sample (number of specimens within the sample),*
- *PIC:POC ratios measured for each sample*
- *The slope of the trendline fitted to the three samples*
- *The R^2 value of each slope*

• Section 3.3: Species names in italics; Syracosphaera needs capitalization.

*25) We will correct this.*

• Fig. 6: Remove grid lines; make lines and symbols the same size and
thickness.

*26) We will update it accordingly.*

• Table 3: Explain acronyms (SS); check digits and only list significant amounts based on the error associated with the calculations/initial measurements. If in doubt, perform error propagation.

27) *We will explain all acronyms in the table caption. We also corrected Table 3, adding the 25% uncertainty adopted around the reported standing sock, living concentration and export concentration. We will explain this in the main text and the caption. The corrected table is added below.*

| Station | Group | Living concentration $C_{living}$ [mg m$^{-3}$] | Standing stock SSm2 [mg m$^{-2}$] | Export concentration $C_{exp}$ [mg m$^{-3}$] |
|---|---|---|---|---|
| 6 | Planktonic gastropod adult | $(5.7 \pm 1.4) \times 10^{-3}$ | $(1.7 \pm 0.4) \times 10^{0}$ | $(3.9 \pm 1.0) \times 10^{-5}$ |
| 6 | Planktonic gastropod juvenile | $(1.8 \pm 0.4) \times 10^{-3}$ | $(5.3 \pm 1.3) \times 10^{-1}$ | $(3.2 \pm 0.8) \times 10^{-4}$ |
| 6 | Planktonic gastropod total | $(7.5 \pm 1.9) \times 10^{-3}$ | $(2.3 \pm 0.6) \times 10^{0}$ | $(3.6 \pm 0.9) \times 10^{-4}$ |
| 6 | Foraminifera > 200um | $(2.1 \pm 0.5) \times 10^{-3}$ | $(3.2 \pm 0.8) \times 10^{-1}$ | $(6.8 \pm 1.7) \times 10^{-5}$ |
| 6 | Foraminifera 125-200 um | $(6.3 \pm 1.6) \times 10^{-7}$ | $(9.5 \pm 2.4) \times 10^{-5}$ | $(2.1 \pm 0.5) \times 10^{-8}$ |
| 6 | Foraminifera total | $(2.1 \pm 0.5) \times 10^{-3}$ | $(3.2 \pm 0.8) \times 10^{-1}$ | $(6.8 \pm 1.7) \times 10^{-5}$ |
| 9 | Planktonic gastropod adult | $(9.8 \pm 2.5) \times 10^{-4}$ | $(2.9 \pm 0.7) \times 10^{-5}$ | 0 |
| 9 | Planktonic gastropod juvenile | $(1.4 \pm 0.4) \times 10^{-3}$ | $(4.3 \pm 1.1) \times 10^{-1}$ | $(1.1 \pm 0.3) \times 10^{-4}$ |
| 9 | Planktonic gastropod total | $(2.4 \pm 0.6) \times 10^{-3}$ | $(7.2 \pm 1.8) \times 10^{-1}$ | $(1.1 \pm 0.3) \times 10^{-4}$ |
| 9 | Foraminifera > 200um | $(1.2 \pm 0.3) \times 10^{-3}$ | $(1.8 \pm 0.5) \times 10^{-1}$ | $(5.9 \pm 1.5) \times 10^{-5}$ |
| 9 | Foraminifera 125-200 um | $(3.6 \pm 0.9) \times 10^{-7}$ | $(5.5 \pm 1.4) \times 10^{-5}$ | $(1.8 \pm 0.4) \times 10^{-8}$ |
| 9 | Foraminifera total | $(1.2 \pm 0.3) \times 10^{-3}$ | $(1.8 \pm 0.5) \times 10^{-1}$ | $(5.9 \pm 1.5) \times 10^{-5}$ |
| 3,4 | Coccolith | not relevant | not relevant | $(1.8 \pm 0.5) \times 10^{-1}$ |
| 3,4 | Coccosphere | $(4.0 \pm 1.0) \times 10^{-2}$ | $(6.9 \pm 1.7) \times 10^{0}$ | $(6.8 \pm 1.7) \times 10^{-3}$ |

**Table 3:** Living concentration ($C_{living}$), integrated standing stock (SSm2) and export concentration ($C_{exp}$) of all plankton groups, separated by station, life stage or size (in case of planktonic gastropods and foraminifera) and shape (in case of coccolithophores). An error of 25% related to measurement uncertainties is assumed around each value.

• Tables (general): Check for consistency, e.g., stdev vs Stdev; check the number of digits (see comment for Table 3); apply consistently across manuscript, tables, and figures.

*28) We checked all other tables in the paper for consistency and number of digits. The corrected version of table 4 and table 5 and included below:*

| Plankton group | Planktonic gastropod | Foraminifera | Coccolith - single | Coccolith - pellet | Coccosphere |
|---|---|---|---|---|---|
| Production (mg m$^{-2}$ day$^{-1}$) | 0.16 ± 0.07 | 0.012 ± 0.004 | Not relevant | Not relevant | 2.0 ± 2.0 |
| $F_{exp}$ (mg m$^{-2}$ day$^{-1}$ | 0.18± 0.05 | 0.019±0.009 | 0.19±0.08 | 24.7±11.2 | 0.03±0.011 |
| production using minimum TT | 0.30 | 0.018 | | | 12 |
| TT calculated (small specimen) | 4±2.7 | 67±89 | Not relevant | Not relevant | Not relevant |
| TT calculated (large specimen) | 40±33 | 17 ± 18 | Not relevant | Not relevant | Not relevant |

**Table 4:** Results of the Monte Carlo simulations for each plankton group, including the standard deviation. Note that especially the calculated turnover times have a very high uncertainty.

| Plankton group | Planktonic gastropod | Foraminifera | Coccolith (single) + coccosphere (single) | Coccolith (pellet) + coccosphere (single) | Coccosphere |
|---|---|---|---|---|---|
| Production (%) | 7 ± 32 | 1 ± 6 | Not relevant | Not relevant | 92 ± 35 |
| Export scenario 1 (%) | 44 ± 12 | 5 ± 3 | 52 ±12 | Not relevant | Not relevant |
| Export scenario 2 (%) | 0.7 ± 9 | 0.08 ± 1 | Not relevant | 99 ± 10 | Not relevant |

**Table 5:** Relative contribution of each plankton group to the production and export of PIC, based on the mean production and export values and their standard deviations calculated using the Monte Carlo simulations (Table 4). We used an additional Monte Carlo simulation to calculate the standard deviation of the percentual contribution of each group. Uncertainties around the estimated contributions are large, related to the large error associated with the measured values as well as the large uncertainties in the sinking speed and turnover time estimates.

- Table 5: Add explanation of export scenario 1 and 2 in the caption.
  *29) We will do this.*

- Figure 7: Same comment as Figure 6. Chlorophyll-a and chlorophyll are used interchangeably; check consistency throughout manuscript.
  *30) We will do this.*

- Fig. 8: Was, I assume, meant to have a Y-axis.
  *31) We will correct this.*

- l. 507 / 508: m. versus m⁻. (consistency); check throughout manuscript, tables, and figures.
  *32) We will do this.*

- l. 507: Capitalize March.
  *33) Thank you for noticing.*

- l. 514–516: This shows that turnover time is vastly overestimated in the case

of pteropods.

*34) As indicated, we will include calculations based on multiple assumptions of turnover in the revised version.*

• l. 667: What is an ashed surface water sample?

*35) We will include a short section describing the plankton pump sampling and include a reference to other studies making use of this pump system to clarify what we mean by 'ashed sample'.*

• Captions of tables in appendix (B2, B3): Species names need italics.

*36) We will correct this.*

• l. 756: pg μm⁻. instead of /μm

*37) We will correct this and check the manuscript for further inconsistencies.*

*Cited literature:*

Alldredge, A. L., & Cohen, Y. (1987). Can Microscale Chemical Patches Persist in the Sea? Microelectrode Study of Marine Snow, Fecal Pellets. Science, 235, 689–691. https://doi.org/10.1126/science.235.4789.689

Anglada-Ortiz, G., Zamelczyk, K., Meilland, J., Ziveri, P., Chierici, M., Fransson, A., & Rasmussen, T. L. (2021). Planktic Foraminiferal and Pteropod Contributions to Carbon Dynamics in the Arctic Ocean (North Svalbard Margin). Frontiers in Marine Science, 8. https://doi.org/10.3389/fmars.2021.661158

Bednaršek, N., Mozina, J., Vogt, M., O'Brien, C., & Tarling, G. A. (2012). The global distribution of pteropods and their contribution to carbonate and carbon biomass in the modern ocean. Earth System Science Data, 4(1), 167–186. https://doi.org/10.5194/essd-4-167-2012

Dong, S., Berelson, W. M., Rollins, N. E., Subhas, A. V., Naviaux, J. D., Celestian, A. J., Liu, X., Turaga, N., Kemnitz, N. J., Byrne, R. H., & Adkins, J. F. (2019). Aragonite dissolution kinetics and calcite/aragonite ratios in sinking and suspended particles in the North Pacific. Earth and Planetary Science Letters, 515, 1–12. https://doi.org/10.1016/j.epsl.2019.03.016

Hagen, E., Feistel, R., Agenbag, J. J., & Ohde, T. (2001). Seasonal and interannual changes in Intense Benguela Upwelling (1982-1999).

Lutjeharms, J. R. E., & Meeuwis, J. M. (1987). The extent and variability of South-East Atlantic upwelling. South African Journal of Marine Science, 5(1), 51–62. https://doi.org/10.2989/025776187784522621

Manno, C., Bednaršek, N., Tarling, G. A., Peck, V. L., Comeau, S., Adhikari, D., Bakker, D. C. E., Bauerfeind, E., Bergan, A. J., Berning, M. I., Buitenhuis, E., Burridge, A. K., Chierici, M., Flöter, S., Fransson, A., Gardner, J., Howes, E. L., Keul, N., Kimoto, K., … Ziveri, P. (2017). Shelled pteropods in peril: Assessing vulnerability in a high CO2 ocean. In Earth-Science Reviews (Vol. 169, pp. 132–145). Elsevier B.V. https://doi.org/10.1016/j.earscirev.2017.04.005

Rogerson, J., Veitch, J., Siedlecki, S., & Fawcett, S. (2025). Frontal features and mixing regimes along the shelf region of the Southern Benguela upwelling system. Continental Shelf Research, 295. https://doi.org/10.1016/j.csr.2025.105560

Siddiqui, C., Rixen, T., Lahajnar, N., Van der Plas, A. K., Louw, D. C., Lamont, T., & Pillay, K. (2023). Regional and global impact of CO2 uptake in the Benguela Upwelling System through preformed nutrients. Nature Communications, 14(1). https://doi.org/10.1038/s41467-023-38208-y

Subhas, A. V., Dong, S., Naviaux, J. D., Rollins, N. E., Ziveri, P., Gray, W., Rae, J. W. B., Liu, X., Byrne, R. H., Chen, S., Moore, C., Martell-Bonet, L., Steiner, Z., Antler, G., Hu, H., Lunstrum, A., Hou, Y., Kemnitz, N., Stutsman, J., … Adkins, J. F. (2022). Shallow Calcium Carbonate Cycling in the North Pacific Ocean. Global Biogeochemical Cycles, 36(5). https://doi.org/10.1029/2022GB007388

Ziveri, P., Gray, W. R., Anglada-Ortiz, G., Manno, C., Grelaud, M., Incarbona, A., Rae, J. W. B., Subhas, A. V., Pallacks, S., White, A., Adkins, J. F., & Berelson, W. (2023). Pelagic calcium carbonate production and shallow dissolution in the North Pacific Ocean. *Nature Communications, 14*(1). https://doi.org/10.1038/s41467-023-36177-w

---

## Author Response (AR1)

Dear associate editor,

Thank you for considering our manuscript for publication in Biogeosciences after minor revisions.

In the corrected manuscript we have addressed all points raised by the reviewers and made the necessary changes as described in detail in the two responses published on the website.

The main concern, that we might overestimate gastropod PIC production by underestimating turnover time, has been dealt with in the following way:

- In section 2.5.1 we added: *The gastropod turnover times adopted in Ziveri at al. (2023) are on the high end of values reported in literature, with several studies reporting turnover times of several months to up to two years (Oakes et al., 2020; Bednaršek et al., 2012; Fabry, 1989). We therefore include an additional calculation of gastropod PIC production, using a longer maximum and minimum turnover time in the Monte Carlo simulation.*
- In Table 2 we added an extra column containing the lower and upper bound (TTmin and TTmax) of the long gastropod turnover scenario, including references. These lower and upper bounds are 183 days and 365 days.
- We rephrased part of section 2.5.3: *We acknowledge that the steady state assumption might not be valid. Pteropods and heteropods are still relatively understudied calcifying plankton groups and especially little is known about their life histories and population dynamics (Bednaršek et al. 2016; Manno et al. 2017; Wall-Palmer et al., 2016), but studies have reported seasonal variation in pteropod and heteropod fluxes in sediment traps (e.g. Oakes et al., 2021, Gardner et al., 2023). However, in the absence of a detailed timeseries of plankton standing stock at our study site we make the simplest assumption at hand.*
- We added the new production results to section 3.4 and Table 5 and 6 (former Table 4 and 5): *Using turnover times of between 0.5 - 1-year results in ~27 times lower gastropod PIC production and a relative contribution of coccolithophores, gastropods and foraminifera of 99, 0.3 and 0.6% respectively.*
- We discuss these new results in section 4.1 and 4.2*:*
  - Lines 558-559: *However, if we adopt a longer gastropod turnover time of 0.5-1 year, the contribution of this group to the PIC production decreases to only 0.3% and coccolithophores dominate production even more, producing 99% of all the PIC.*
  - Lines 575-580: *Gastropod production and export balance when using the short turnover times to calculate production. When we assume turnover times to lie between 0.5-1 year, export is 30 times higher than production.*

*This suggests that at our study site gastropod turnover is faster than 0.5-1 year, or that gastropod export concentration or sinking speeds are overestimated. A recent review paper by Ziveri et al. (2025) also finds generally higher gastropod export fluxes than production estimates and suggest that adopting lower gastropod turnover times, on the order of a few weeks instead of 1 year, could bring these values closer together.*

- A small adjustment has been made to the last lines of the abstract: *Coccolithophores contributed 92% - 99% of the produced PIC, depending on planktonic gastropod turnover time, and from 52 to 99% of the exported PIC, depending on their mode of sinking. Both the standing stock and export of planktonic gastropods was significantly larger than that of foraminifera. Similarity between our results and those from different ocean basins suggests that these patterns are global in nature, implying that not only coccolithophores but also gastropods may be a more important contributor to the oceans PIC inventory than foraminifera, challenging a longstanding paradigm.*

Additionally, we have made the following corrections:

- We have replaced Figure 5 by a table (Table 3). This means that the table and figure numbering has changed from there on. Figure 5 has instead been added to the Appendix as figure B1.
- For brevity and readability, we have changed 'planktonic gastropod' to 'gastropod' throughout the entire manuscript, except for the Abstract and Introduction. In the Introduction, we introduce the planktonic gastropods and mention that from thereon, they will be referred to as 'gastropods.
- We have changed the word 'multinet' to 'MultiNet' throughout the manuscript.
- We clarified the use of station 39 samples in section 2.2.1, 'Measuring PIC/POC ratio of selected gastropod species': *We strove to use measured rather than calculated PIC mass where possible. The pteropod species Limacina bulimoides and Heliconoides inflatus, occurring in high abundances in the surface nets, were processed separately from the bulk gastropod samples to obtain species-specific PIC/POC ratios. For this purpose, we used individuals collected in net 5 at stations 6 and 9, as well as specimens from net 5 at station 39, located further north. The inclusion of the station 39 material enlarges the sample size on which we base our species-specific PIC/POC ratio estimate. This approach requires the assumption that the more northerly position of station 39 does not introduce a systematic latitudinal bias in PIC/POC ratios for these species. We will compare our species-specific PIC/POC ratios to those reported by Bednaršek et al. (2012). We additionally calculate average PIC ind$^{-1}$ and POC ind$^{-1}$ based on stations 6 and 9 only and use those in our own study to reconstruct the PIC mass of L. bulimoides and H. inflatus in the unweighed nets, to stay as close as possible to our site-specific measurements.*

- We corrected equation 4. The equation represents the calculation of gastropod PIC concentration in the productive zone, and should be, as the text correctly describes, 'total PIC mass in the upper 300 m divided by the total amount of water filtered by the three nets'. However, in the previous version, the division by volume was missing in the equation. We have corrected this, and the full equation now reads:

$$C_{bpz-0} = (MassPIC_{net3} + MassPIC_{net4} + MassPIC_{net5})/(V_{net3} + V_{net4} + V_{net5}$$

  The calculations in the R-script already used the correct version of the equation, so no changes were necessary there.

- In our reply to Dr. Keul, we wrote that we would add a short line to the methods section to clarify that reconstructed weights and concentrations were corrected for the splitting of the samples. We decided not to add more information to the main text but clarify this in Appenix A, line 675: *To obtain the total CaCO$_3$ weight of each sample, the original counted number of full and empty shells was first multiplied by 2, to correct for the splitting of the sample, and then multiplied by this average shell CaCO$_3$ weight (W$_{shell}$).*

- In Table 5 (former Table 4) we changed the row '*Production using minimum TT' to 'Production using minimum coccolithophore TT'* and presented only the production value based on the coccosphere standing stocks. We changed this because the gastropod and foraminifera values calculated using only the minimum turnover estimate are not discussed anywhere in the manuscript and thus are not relevant here.

- We added some text to line 781, in which we explicitly state that all raw data and calculations related to the PIC/POC ratios of selected gastropods from station 6, 9 and 39 can be found on GitHub and Zenodo. An additional data file has been added to GitHub and a new release of Zenodo has been made. The reference to the Zenodo release has been updated in the bibliography. The additional data file does not contain new data, but provides the calculations that led from raw counts and mass measurements to the PIC/POC ratio of each of the selected gastropod groups at stations 6, 9 and 39.

We hope to have adequately addressed all issues and look forward to hearing your reply.

Yours sincerely,

Anne Kruijt, on behalf of all authors.